# Porous hypercrosslinked polymer-TiO$_2$-graphene composite photocatalysts for visible-light-driven CO$_2$ conversion

Shaolei Wang[1], Min Xu[1], Tianyou Peng[2], Chengxin Zhang[1], Tao Li[1], Irshad Hussain [3], Jingyu Wang [1] & Bien Tan [1]

Significant efforts have been devoted to develop efficient visible-light-driven photocatalysts for the conversion of CO$_2$ to chemical fuels. The photocatalytic efficiency for this transformation largely depends on CO$_2$ adsorption and diffusion. However, the CO$_2$ adsorption on the surface of photocatalysts is generally low due to their low specific surface area and the lack of matched pores. Here we report a well-defined porous hypercrosslinked polymer-TiO$_2$-graphene composite structure with relatively high surface area i.e., 988 m$^2$ g$^{-1}$ and CO$_2$ uptake capacity i.e., 12.87 wt%. This composite shows high photocatalytic performance especially for CH$_4$ production, i.e., 27.62 μmol g$^{-1}$ h$^{-1}$, under mild reaction conditions without the use of sacrificial reagents or precious metal co-catalysts. The enhanced CO$_2$ reactivity can be ascribed to their improved CO$_2$ adsorption and diffusion, visible-light absorption, and photo-generated charge separation efficiency. This strategy provides new insights into the combination of microporous organic polymers with photocatalysts for solar-to-fuel conversion.

[1] Key Laboratory of Material Chemistry for Energy Conversion and Storage (Ministry of Education), Hubei Key Laboratory of Material Chemistry and Service Failure, School of Chemistry and Chemical Engineering, Huazhong University of Science and Technology, Luoyu Road No. 1037, 430074 Wuhan, China. [2] College of Chemistry and Molecular Science, Wuhan University, Bayi Road No. 299, 430072 Wuhan, China. [3] Department of Chemistry & Chemical Engineering, SBA School of Science & Engineering, Lahore University of Management Sciences (LUMS), DHA, Lahore Cantt, Lahore 54792, Pakistan. These authors contributed equally: Shaolei Wang, Min Xu. Correspondence and requests for materials should be addressed to J.W. (email: wangjingyu@hust.edu.cn) or to B.T. (email: bien.tan@mail.hust.edu.cn)

The rapid consumption of carbon-rich fossil fuels has accelerated global energy shortage and significantly increased the $CO_2$ emissions causing serious environmental issues including greenhouse effect responsible for global warming. Among various strategies for $CO_2$ conversion, the photoreduction of $CO_2$ into chemical fuels has attracted increasing attention recently because it utilizes the abundant and sustainable solar energy to mimic the natural photosynthesis[1,2]. So far, two approaches, i.e., homogeneous system and heterogeneous system, have been developed for the photoreduction of $CO_2$. In homogeneous systems, the molecular metal complexes have shown high photocatalytic reactivity towards $CO_2$ reduction[3–5]. In contrast, the conversion efficiency in heterogeneous systems is relatively low, and the exploration of high performance heterogeneous photocatalysts is highly desired keeping in view their higher stability and recyclability[6,7].

In heterogeneous photocatalytic systems, the electrons are generated by light absorption and then transferred to the catalytically active sites to react with the adsorbed $CO_2$ molecules, so the conversion efficiency essentially relies on the light absorption ability, generation and separation of the photogenerated charge carriers, and $CO_2$ adsorption and diffusion[7,8]. Tremendous efforts have been made to optimize the structure and composition of semiconductor photocatalysts to improve their visible-visible light absorption and charge separation efficiency, e.g., constructing heterojunctions, creating surface defects, introducing metal co-catalysts, and engineering exposed crystal facets etc[9–14]. The $CO_2$ adsorption ability is particularly crucial to photocatalytic heterogeneous systems for $CO_2$ conversion, which readily occurs at the active sites of photocatalysts developing intimate contact with $CO_2$ molecules[1,7,15]. Unfortunately, the $CO_2$ adsorption on the surface of semiconductor photocatalysts is extremely low due to their low specific surface area and the lack of matched pores[7,16]. To overcome this limitation, the researchers have modified the photocatalytic reaction systems by elevating pressure, adding sacrificial reagents, or introducing $CO_2$-philic solvents[13,17,18]. In contrast, it is relatively difficult to achieve efficient photocatalytic reduction of $CO_2$ under mild gas–solid reaction conditions without the use of sacrificial reagent or precious metal co-catalyst, and can be further enhanced by the rational design of the microporous structure to facilitate the $CO_2$ uptake and conversion.

Given this challenge, incorporating a $CO_2$ capture material into the photocatalytic system has great potential to provide an opportunity for improving $CO_2$ conversion efficiency. For example, the integration of metal-organic frameworks (MOFs)

with photocatalysts has been demonstrated to offer better adsorptive sites for gas uptake because of their larger surface area and microporosity[19–21]. Although the $CO_2$ conversion efficiency has been greatly improved by MOFs incorporation (Supplementary Table 1), the photocatalytic performance is still not sufficient for the practical applications, especially because the methane generation is quite limited[22]. It has been established that the prerequisites to $CO_2$ transformation involve two steps i.e., $CO_2$ capture and diffusion to the catalytic sites[7,15]. The porous capture materials possess abundant adsorptive sites but they are less catalytically active for $CO_2$ reduction than semiconductors or precious metals[23]. Thus the photoreduction efficiency largely depends on the $CO_2$ diffusion from the capture materials to the photocatalysts[19]. Therefore, in order to achieve higher $CO_2$ conversion, an efficient photocatalyst requires high $CO_2$ uptake as well as short diffusion length.

As a proof of concept, we develop a porous composite structure by in situ knitting hypercrosslinked polymers (HCPs) on $TiO_2$-functionalized graphene ($TiO_2$-FG). The HCPs materials as pure organic microporous materials show large surface area, high $CO_2$ uptake, and excellent physicochemical stability[24]. It is worth mentioning that this is the example involving the combination of microporous organic polymers with photocatalysts for $CO_2$ conversion among the numerous reported photocatalysts. The synthetic strategy of such well-defined porous composite structure is depicted in Fig. 1. The $TiO_2$-G composite is initially obtained by the reduction of graphene oxide (GO) followed by in situ growth of the anatase $TiO_2$ crystals with reactive {001} facets on its surface by a solvothermal process[25,26]. The graphene skeleton of $TiO_2$-G composite is functionalized to form $TiO_2$-FG and provide an open phenyl group for covalent linking (step I). Finally, the ultrathin polymer layers are hypercrosslinked on $TiO_2$-FG by the knitting of syn-PhPh$_3$ and the open phenyl groups on graphene (step II), resulting in the formation of the porous hypercrosslinked polymer-$TiO_2$-graphene (HCP-$TiO_2$-FG) composite[27,28]. Due to the enhanced $CO_2$ adsorption ability of HCPs and the short diffusion length around the $TiO_2$ photocatalysts, such well-defined HCP-$TiO_2$-FG structure is envisioned to enhance the reactivity of $CO_2$ molecules to facilitate the production of $CH_4$.

## Results

### Characterization of the resulting materials

The morphology and microstructure of the as-prepared materials were investigated by transmission electron microscopy (TEM), field-emission

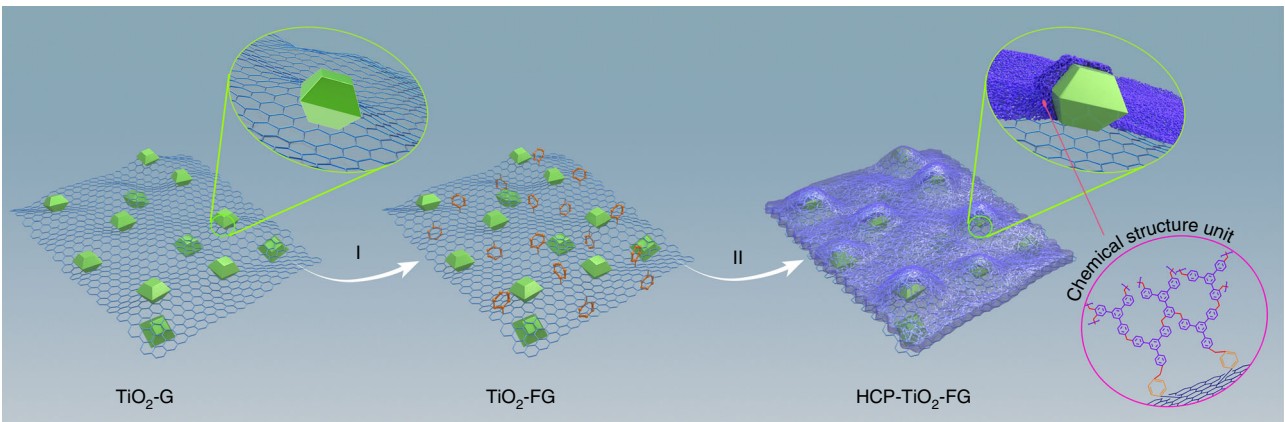

**Fig. 1** Construction of a well-defined porous HCP-$TiO_2$-FG composite structure. I The functionalization of $TiO_2$-G by diazonium salt formation. II The knitting of $TiO_2$-FG with syn-PhPh$_3$ by solvent knitting method. The magnified model in the top right corner is the cross profile of HCP-$TiO_2$-FG composite

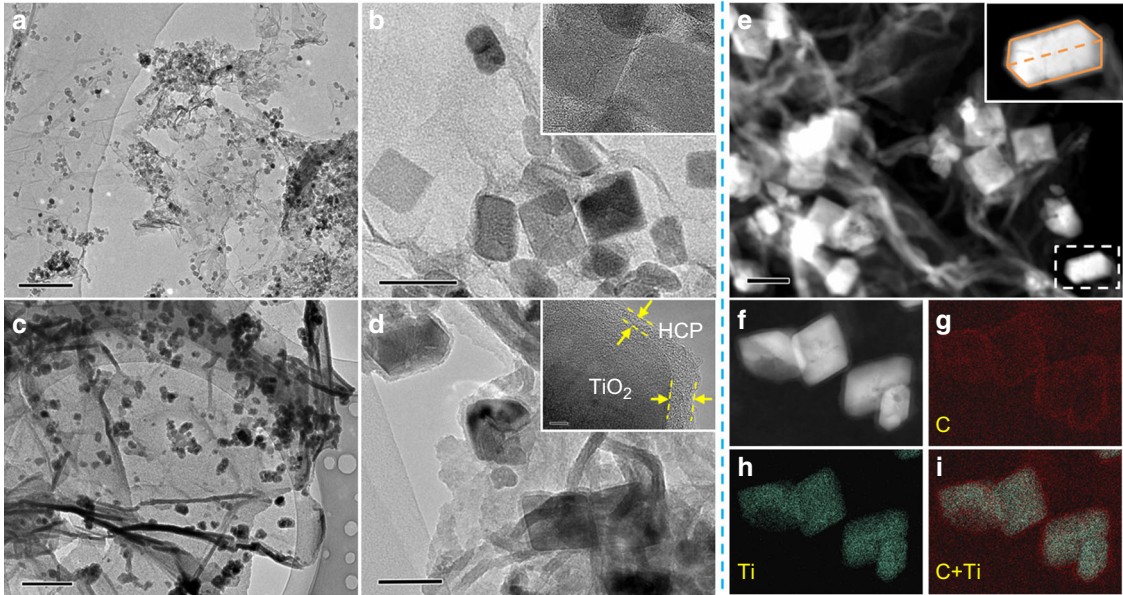

**Fig. 2** Morphology and elemental mapping of various photocatalysts. TEM images of **a, b** $TiO_2$-G and **c, d** HCP-$TiO_2$-FG at different magnification. The insets in **b** and **d** are the corresponding HR-TEM images. **e** STEM image of HCP-$TiO_2$-FG. **f–i** High-angle annular dark field (HAADF) mapping images of HCP-$TiO_2$-FG. The scale bar are 200 nm in **a**, 50 nm in **b**, 0.5 μm in **c**, 100 nm in **d**, 100 nm in **e**, and 5 nm in insets in **b** and **d**

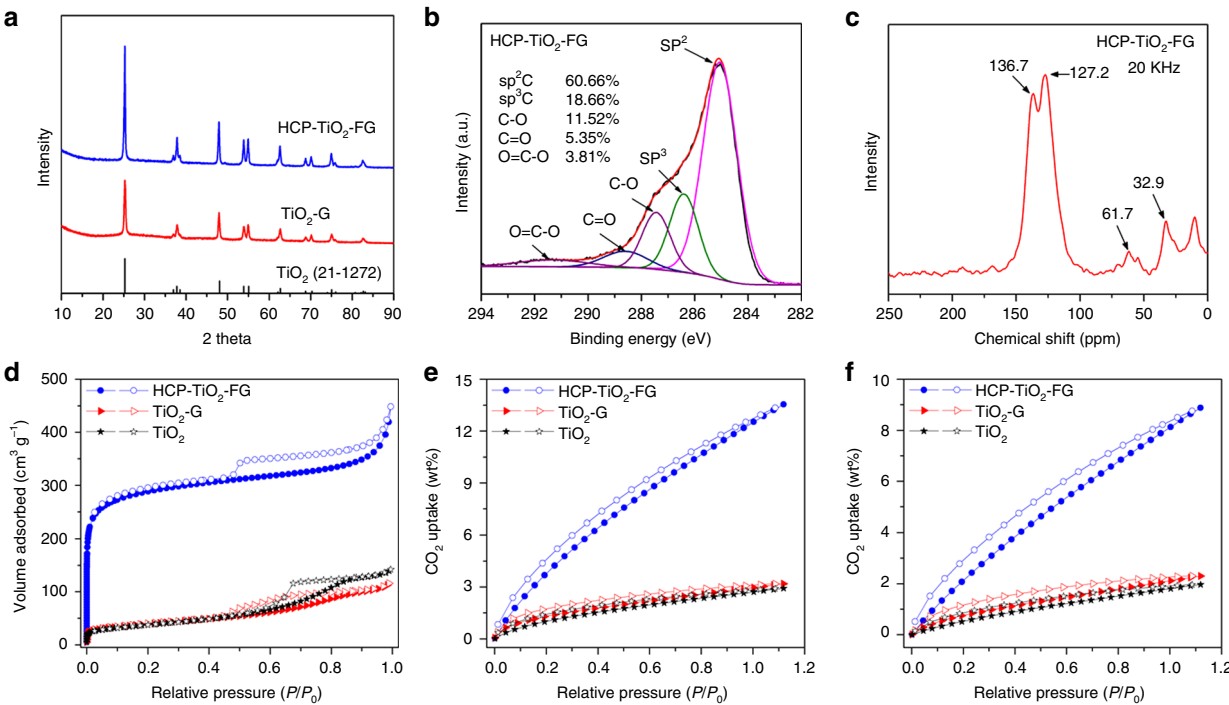

**Fig. 3** Chemical structure, porosity, and $CO_2$ uptake of various photocatalysts. **a** XRD image of $TiO_2$, $TiO_2$-G, and HCP-$TiO_2$-FG. **b** C [1s] profiles of HCP-$TiO_2$-FG. **c** [13]C cross-polarization/magic-angle spinning (CP/MAS) NMR spectra of HCP-$TiO_2$-FG. **d** Nitrogen adsorption and desorption isotherms at 77.3 K of samples. **e** Volumetric $CO_2$ adsorption isotherms and desorption isotherms up to 1.00 bar at 273.15 K of samples. **f** Volumetric $CO_2$ adsorption isotherms and desorption isotherms up to 1.00 bar at 298.15 K of samples

scanning electron microscopy (FE-SEM), and atomic force microscopy (AFM). FE-SEM characterization of pure HCPs from our previous knitting method showed a layered bulk structure[27,28], whereas the HCP-$TiO_2$-FG exhibited a 3D morphology with flake-like graphene sheets, which is similar to the reported porous graphene-based materials[29,30]. The absence of naked $TiO_2$ crystals on the graphene surface indicated their complete coating by the HCPs layers in HCP-$TiO_2$-FG (Supplementary Figure 1a–d). The

TEM characterization further showed that the smooth graphene nanosheets (Supplementary Figure 2) were uniformly decorated with $TiO_2$ crystals to confirm the formation of $TiO_2$-G composite nanostructure (Fig. 2a,b). After in situ knitting, a distinct composite structure was formed of HCP-$TiO_2$-FG in which the graphene surface and $TiO_2$ crystals were covered by the HCPs layers. No freestanding HCPs blocks were observed in SEM, TEM, and scanning transmission electron microscopy (STEM) images

(Fig. 2c–e and Supplementary Figure 1d). High-resolution TEM (HR-TEM) characterization showed that the $TiO_2$ crystals were fully wrapped by an ultrathin HCPs layer with a thickness of 3–8 nm (Fig. 2d). The typical AFM observation and thickness analysis revealed a uniform thickness of HCP-$TiO_2$-FG as 10 ± 0.5 nm, whereas the thickness of $TiO_2$-FG was only 4 ± 0.5 nm suggesting the formation of composite structure (Supplementary Figures 3–4). The percentage of the exposed {001} facets in the $TiO_2$ crystal was calculated to be ~30% using a geometric calculation (Supplementary Figure 5). The elemental mapping images in Fig. 2f–i clearly display the thin HCP shells wrapping the surface of $TiO_2$ crystals. By rotating the angle of the sample, multiple images were collected to create a three-dimensional TEM (3D-TEM) movie (Supplementary Figure 6 and Supplementary Movie 1) to further elucidate the HCP-$TiO_2$-FG composite structure with distinct interface between $TiO_2$ and HCP-FG. Based on the above analysis, it can be deduced that the $TiO_2$ crystals were supported on the graphene sheets and then encapsulated by the ultrathin HCPs layers after knitting syn-$PhPh_3$ with functionalized graphene, as shown in Fig. 1.

The X-ray diffraction (XRD) pattern showed that all the samples consisted of pure anatase $TiO_2$ crystals. The introduction of HCPs layers did not alter the crystal phase of $TiO_2$ but caused an obvious increase in the particle size (Fig. 3a), which followed the Ostwald ripening mechanism during long-time refluxing in the functionalization and knitting processes. The composition and surface chemical structure were investigated by X-ray photoelectron spectroscopy (XPS) measurements. Unlike $TiO_2$-G, the peak intensities of Ti and O signals for $TiO_2$-FG and HCP-$TiO_2$-FG were gradually weakened due to the decrease of $TiO_2$ content (Supplementary Figure 7a). As shown in high-resolution $C^{1s}$ spectra, the ratio of $sp^2$ C and $sp^3$ C signals demonstrated an increasing trend after functionalization and knitting, which was attributed to the introduction of more $sp^2$ C compared with the formation of $sp^3$ C (Fig. 3b and Supplementary Figure 7b,c)[30]. Interestingly, the location of $Ti^{2p}$ displayed an obvious shift of ~0.2 eV towards higher energy after HCPs layers formation (Supplementary Figure 7d). This shift verified the electronic interaction of HCPs layers with $TiO_2$, which was favorable for the electron transfer at the heterojunction interface. Fourier transform infrared (FT-IR) spectroscopy was carried out to investigate the chemical structure of the resulting materials. Compared with the FT-IR spectrum of $TiO_2$-FG, strong C–H stretching vibrations of methylene near $2920\ cm^{-1}$ and aromatic ring skeleton vibration peaks near $1485\ cm^{-1}$ are clearly visible for HCP-$TiO_2$-FG (Supplementary Figure 8)[27,28]. The slight shift of the Ti-O-Ti stretching vibration further verified the interfacial interactions of $TiO_2$ with HCPs layers in the composite materials.

The $^{13}$C cross-polarization/magic-angle spinning nuclear magnetic resonance (CP/MAS NMR) was employed to further confirm the proposed functionalization and knitting processes at the molecular level. The introduction of phenyl groups in $TiO_2$-FG resulted in the appearance of a shoulder peak arising at 136.7 ppm in the aromatic carbon region due to the functionalization of $TiO_2$-G, and the resonance at 127.2 ppm are assigned to the $sp^2$ carbons of the graphene based on reported data (Supplementary Figure 9)[30]. The enhanced intensity of resonance peaks near 136.7 ppm can be ascribed to the introduction of abundant $sp^2$ carbon by knitting syn-$PhPh_3$ with $TiO_2$-FG for HCP-$TiO_2$-FG. Meanwhile, the methylene linkers formed by knitting processes resulted in the appearance of a new peak near 32.9 ppm (Fig. 3c)[27,28]. The $TiO_2$ content and thermostability of as-prepared materials were investigated by thermogravimetric analysis (TGA). As expected, the percent weight loss of $TiO_2$ significantly decreased from the initial 67% for $TiO_2$-G to 31% for HCP-$TiO_2$-FG suggesting the incorporation of HCPs layers into the composite (Supplementary

Figure 10). The results are in good agreement with the accurate measurement by inductively coupled plasma-mass spectrometry (ICP-MS) analysis. More importantly, the HCP-$TiO_2$-FG composite structure exhibited the excellent thermal stability comparable to $TiO_2$-G with resistance to degradation up to 400 °C, presumably due to the formation of HCPs layers on rigid graphene skeletons.

**Porosity and $CO_2$ uptake of the resulting materials**. After confirming the morphology and chemical structure of the as-prepared materials, we further investigated their porosity parameters as shown in Fig. 3d and Supplementary Table 2. The $TiO_2$ and $TiO_2$-FG showed type IV isotherms with a minute amount of adsorbed nitrogen and obvious hysteresis loops at medium pressure region, indicating low surface area and the existence of mesopores[31,32]. The isotherms of HCP-$TiO_2$-FG exhibited a type I character with a steep nitrogen gas uptake at low relative pressure ($P/P_0 < 0.001$) thus reflecting abundant microporous structure. The existence of an obvious hysteresis and a slight rise at medium and high pressure region revealed the presence of mesopores and macropores, respectively[33]. More importantly, the introduction of HCPs layers dramatically enlarged the specific surface area of $TiO_2$-G to $988\ m^2\ g^{-1}$, together with an increase in micropore volume from 0.009 to $0.306\ cm^3\ g^{-1}$, which is, in fact, much higher than the reported semiconductor-graphene composites or most of the porous photocatalysts (Supplementary Table 3). Moreover, the dominant pore diameter of HCP-$TiO_2$-FG was centred at about 0.5 and 1.1 nm, along with continuous mesoporous and macroporous structure (Supplementary Figure 11). The high specific surface area and abundant ultra-microporous nature of the HCP-$TiO_2$-FG composite inspired us to investigate its gas uptake capacity. It is well established that the $CO_2$ uptake by porous polymer materials mainly results from its physical adsorption[34,35]. Such adsorption mode displays the pressure and temperature-dependent features with excellent recyclability for the repeated $CO_2$ adsorption and desorption (Supplementary Figure 12). Interestingly, the $CO_2$ uptake of the HCP-$TiO_2$-FG reached as high as 12.87 wt% at 1.00 bar and 273.15 K, which is more than four-fold higher than those of $TiO_2$ and $TiO_2$-G. These results were finally compared with some porous photocatalysts reported under similar conditions (Fig. 3e,f and Supplementary Table 3). It was found that the $CO_2$ uptake of HCP-$TiO_2$-FG was higher than that of HCP-FG (Supplementary Table 2), implying that the $TiO_2$ intercalation somewhat restricted the aggregation of HCPs layers on graphene nanosheets. To further understand the effect of such well-defined HCP-$TiO_2$-FG composite structure on improving surface area and $CO_2$ uptake, another type of composite, $TiO_2$/HCP-FG, was prepared as a control by changing the order of introducing HCP and $TiO_2$. The HCP layers were hypercrosslinked on the functionalized graphene to form HCP-FG at first, and then $TiO_2$ crystals were grown on the HCP-FG surface during the solvothermal process. Owing to the high surface area and porous property of HCP-FG support, the $TiO_2$ particles of $TiO_2$/HCP-FG possessed much smaller size than that of $TiO_2$-G or HCP-$TiO_2$-FG (Supplementary Figure 13). However, the hypercrosslinking reaction caused the graphene surface to be almost entirely covered by HCP layers, so that most of $TiO_2$ crystals were assembled on the HCP surface rather than be encapsulated by HCPs layers like those in HCP-$TiO_2$-FG composite. The results implied the interaction of $TiO_2$ with graphene serving as bridge to the formation of well-defined HCP-$TiO_2$-FG composite structure. The $TiO_2$/HCP-FG composite showed a high surface area of $178\ m^2\ g^{-1}$ and $CO_2$ uptake of 3.31 wt% relative to $TiO_2$ and $TiO_2$-G (Supplementary Figure 14), but much lower than that of HCP-$TiO_2$-FG. According to the previous such reports, the incorporation of semiconductor

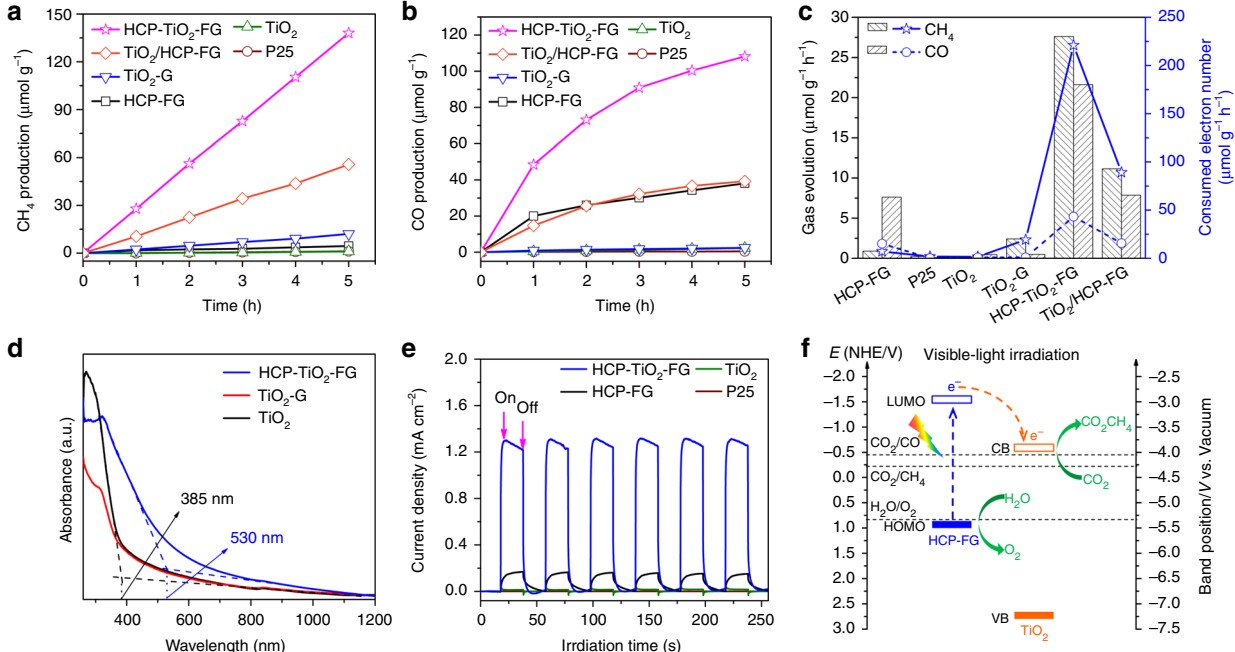

**Fig. 4** The photocatalytic performance of $CO_2$ reduction, optical and photoelectrical properties, and mechanism of charge transfer pathway. Time-dependent production of **a** $CH_4$ and **b** CO in photocatalytic $CO_2$ reduction with different catalysts under visible-light ($\lambda \geq 420$ nm). The photocatalytic reactions were carried out in a batch system under standard atmospheric pressure. The partial pressure of $CO_2$ and $H_2O$ were constant with the water contents below the scaffold loading photocatalyst. Under visible-light irradiation, the temperature of the water was measured to be about 50 °C. **c** Average efficiency of photocatalytic $CO_2$ conversion with different catalysts during 5 h of visible-light ($\lambda \geq 420$ nm) irradiation. **d** UV-Vis absorption spectra of $TiO_2$, $TiO_2$-G, and HCP-$TiO_2$-FG catalysts. **e** Amperometric $I-t$ curves of samples under visible-light ($\lambda \geq 420$ nm) irradiation. **f** Proposed mechanism of charge separation and transfer within the HCP-$TiO_2$-FG composite photocatalyst under visible-light ($\lambda \geq 420$ nm) irradiation

| **Table 1 Summary of the photocatalytic $CO_2$ conversion efficiency during 5 h of constant irradiation** | | | | | |
|---|---|---|---|---|---|
| **Photocatalyst** | **Visible-light irradiation ($\mu mol\ g^{-1}\ h^{-1}$)** | | | | |
| | $r(CH_4)^a$ | $r(CO)^b$ | $R_e(CH_4)^c$ | $R_e(CO)^d$ | $R_e^e$ |
| P25 | 0.23 | 0.10 | 2 | 0 | 2 |
| $TiO_2$ | 0.21 | 0.46 | 2 | 1 | 3 |
| $TiO_2$-G | 2.42 | 0.49 | 19 | 1 | 20 |
| HCP-FG | 0.90 | 7.62 | 7 | 15 | 22 |
| $TiO_2$/HCP-FG | 6.71 | 16.17 | 54 | 32 | 86 |
| HCP-$TiO_2$-FG | 27.62 | 21.63 | 221 | 43 | 264 |

a,bAverage of gas evolution rate ($r$) during 5 h of photocatalytic $CO_2$ reduction
c,dRate of electron consumption for $CH_4$ and CO evolution; $R_e(CH_4) = 8r(CH_4)$, $R_e(CO) = 2r$ (CO)
e$R_e$ is the rate of total consumed electron number for the reduced product; $R_e = 8r(CH_4) + 2r$ (CO)
fP25 is the commercial $TiO_2$ (Degussa)

photocatalysts generally decrease the surface area and $CO_2$ uptake of the capture materials, mainly resulting from the semiconductors with low surface area occupying the porous surface[19,23,36,37]. Hence the superiority of HCP-$TiO_2$-FG in $CO_2$ adsorption could be ascribed to the well-defined porous composite structure with $TiO_2$ encapsulated inside the HCP-FG network instead of being assembled on the surface.

**Photoreduction $CO_2$ activity of the resulting materials.** Given ideal pore distribution and excellent $CO_2$ capture capacity of HCP-$TiO_2$-FG, we set out to evaluate the photocatalytic efficiency towards $CO_2$ conversion in a gas–solid reaction system. The photoreduction of $CO_2$ proceeded under mild conditions without

any photosensitizer or organic sacrificial reagent. Figure 4a,b and Table 1 show the yield of the $CO_2$ conversion products during 5 h of photocatalytic reaction under visible-light (wavelength $\lambda \geq 420$ nm) irradiation. The CO and $CH_4$ gases were generated as main products via the two-electron and eight-electron reduction processes, respectively. The porous HCP-$TiO_2$-FG catalyst presented high average conversion efficiency with a rate of total consumed electron number ($R_e$) as 264 $\mu mol\ g^{-1}\ h^{-1}$, a $CH_4$ evolution rate of 27.62 $\mu mol\ g^{-1}\ h^{-1}$, and a CO evolution rate of 21.63 $\mu mol\ g^{-1}\ h^{-1}$. Figure 4c shows the dominant electron consumption selectivity for $CH_4$ production as high as 83.7%. More importantly, no $H_2$ evolution was detected during the photocatalytic reaction suggesting that the HCP-$TiO_2$-FG material possessed the high selectivity for photoreduction of $CO_2$ and effectively hindered the side reaction of $H_2O$ reduction. To the best of our knowledge, these $CO_2$ photoconversion results are the best among those of the recently reported heterogeneous photocatalysts under similar gas–solid reaction conditions (with no sacrificial reagent), especially much higher than those of photocatalysts without precious metal co-catalysts (Supplementary Table 1)[11,17,23,38–40].

The well-defined HCP-$TiO_2$-FG composite photocatalyst with abundant microporosity demonstrated excellent performance for $CO_2$ reduction but its underlying mechanism still needs to be investigated. The $CO_2$ conversion products were rarely detected (<1 $\mu mol\ g^{-1}\ h^{-1}$) over commercial $TiO_2$ (P25), and pure $TiO_2$ with reactive {001} facets due to their limited light-responsive ability in the visible region (Fig. 4d). Coupling of $TiO_2$ with graphene to produce $TiO_2$-G obviously increased the $CH_4$ production (2.42 $\mu mol\ g^{-1}\ h^{-1}$) by improving visible-light absorption and electron transport property[18,41]. The adsorptive and catalytic sites can be clarified through the comparison in porous property and photocatalytic performance. Obviously, the

introduction of porous HCPs layers enriched the adsorptive sites to achieve the high $CO_2$ uptake and improved the visible light absorption. Thus the formation of well-defined HCP-$TiO_2$-FG composite structure resulted in much higher photocatalytic $CO_2$ reduction rate. The HCP-FG material also exhibited broad visible-light absorption, high surface area, and notable $CO_2$ uptake (Supplementary Table 2). However, its photocatalytic performance was far less than that of HCP-$TiO_2$-FG, especially in the eight-electron reduction to $CH_4$ (Fig. 4a and Table 1). It is well known that the polymer materials usually possess the excitons with high binding energy, which usually recombine at the excited states[42]. That is, the adsorptive sites on the porous capture material are generally catalytically inactive for $CO_2$ reduction[23]. To further elucidate the superiority of porous HCP-$TiO_2$-FG composite, the $CO_2$ conversion efficiency over $TiO_2$/HCP-FG composite was evaluated. It was found that the photocatalytic activity of $TiO_2$/HCP-FG was much lower than that of HCP-$TiO_2$-FG. It may be due to the blockage of the porous structure of HCP-FG with $TiO_2$ crystals thereby decreasing the surface area and $CO_2$ uptake and subsequently leading to the reduced $CO_2$ reduction rate. Although $TiO_2$ deposition blocked most of the adsorptive sites of HCP-FG and resulted in a dramatic decrease to less than one-third of $CO_2$ uptake, the $CH_4$ production over $TiO_2$/HCP-FG was, however, 7.4 times more than that over pristine HCP-FG. The comparison among HCP-FG, HCP-$TiO_2$-FG, and $TiO_2$/HCP-FG shows that the catalytic sites on $TiO_2$ are much more active for $CO_2$ reduction than those on HCP-FG. The model of $CO_2$ diffusion and conversion is presented in Supplementary Figure 15. Based on the above analysis, it can be deduced that the in situ knitting strategy for HCP-$TiO_2$-FG can effectively produce porous structure without significant pore blockage of the porous polymers. More importantly, the HCPs obtained by this strategy are comprised of ultrathin layers with a thickness of 3–8 nm wrapping around $TiO_2$ crystals (Fig. 2d–f), which facilitates the diffusion of $CO_2$ molecules from the adsorptive sites on HCPs layers to the catalytic sites on $TiO_2$ photocatalysts. The effect of the thickness of HCP layers on the $CO_2$ conversion efficiency was studied by adjusting the amount of syn-PhPh3. By increasing the amount of syn-PhPh3, the mass ratio of $TiO_2$ was slightly decreased from 31 to 29% (Supplementary Figure 16), however, the size of $TiO_2$ particles was decreased accompanied by the thickening of the HCP layers (Supplementary Figure 17), which suggests that the HCP outer layers effectively suppress the growth of $TiO_2$ crystals. The distinct thickening of the outer layers was further verified from the characteristic morphology revealed in Fig. 1, showing HCP layers being hypercrosslinked on FG surface and encapsulating $TiO_2$ crystals. The surface area and $CO_2$ uptake capacity increased with the amount of syn-PhPh3 (Supplementary Figure 18a and Supplementary Table 4), on the other hand, the diffusion length of $CO_2$ molecules also increased due to the thickening of the outer layer. As the $CO_2$ conversion efficiency increased initially and then decreased at higher amount (Supplementary Figure 18b), there may be an appropriate thickness of HCP layers that balance the $CO_2$ adsorption and diffusion. As a result of the relatively high photocatalytic performance of porous HCP-$TiO_2$-FG, the $O_2$ evolution can be measured to provide the evidence of the oxidation cycle offering a better insight of the mechanism that is seldom discussed in the literature[43]. The $O_2$ evolution rate over HCP-$TiO_2$-FG under visible-light irradiation was determined to be 1.6 μmol h$^{-1}$, while the $O_2$ evolution over other photocatalysts was too low to be detectable (Supplementary Figure 19). The electrons from the water oxidation are slightly higher than the total consumed electrons for the reduced products including $CH_4$ and CO.

To verify the evolution of CO and $CH_4$ from $CO_2$ conversion over HCP-$TiO_2$-FG photocatalyst, we conducted three controlled experiments: (1) irradiation of catalyst under inert $N_2$ condition; (2) the use of isotopically labeled $^{13}CO_2$ and $H_2^{18}O$ as the reactants; (3) irradiation of catalyst in the presence of $CO_2$ gas without $H_2O$ vapors. Under inert $N_2$ condition, no $CH_4$ was detected and the CO yield was only 7% compared to that under $CO_2$ atmosphere (Supplementary Figure 20). The trace CO product might be generated by the decomposition of the residual oxygen-containing functional groups of graphene, which was evidenced by XPS analysis (Supplementary Figure 7a). In an isotopically labeled experiment, the $^{13}CH_4$ and $^{13}CO$ signals at $m/z = 17$ and $m/z = 29$ appeared after the photocatalytic reaction. The results confirmed that the CO and $CH_4$ products are indeed originating from the photocatalytic reduction of $CO_2$ gas (Supplementary Figure 21). The isotopically labeled $H_2^{18}O$ vapors led to the formation of $^{18}O_2$ (Supplementary Figure 22), suggesting that the evolved $O_2$ gas was derived from the photocatalytic water oxidation. In the absence of $H_2O$ vapors, the $CH_4$ evolution was rapidly declined to ~9% of the original rate, while ~1.8 times increase was observed in CO evolution, revealing that the $H_2O$ vapors act as the proton donors for the conversion of $CO_2$ to $CH_4$[44]. Although the porous HCP-$TiO_2$-FG also exhibits a high adsorption capacity towards water vapors, about 30 wt% at 90% humidity (Supplementary Figure 23), the existence of water vapors brings a slight increase in $CO_2$ uptake (Supplementary Figure 24), presumably due to their affinity with the water molecules. The $CH_4$ evolution is relatively difficult since the reactivity of the adsorbed $CO_2$ molecules should be high enough to accept eight electrons and eight protons to break the C–O bonds and form the C–H bonds[45]. Normally, the adsorbed $CO_2$ molecules are more readily converted to CO than $CH_4$ on the surface of semiconductor photocatalysts[18,19,36]. Precious metal co-catalysts are generally introduced to improve the reactivity of $CO_2$ molecules to obtain more $CH_4$ production, e.g., 19.6 μmol g$^{-1}$ h$^{-1}$ over Pd7Cu1-loaded $TiO_2$ and 20.6 μmol g$^{-1}$ h$^{-1}$ over $TiO_2$-PdH0.43 under UV-light[40,46]. However, the $CH_4$ evolution over the HCP-$TiO_2$-FG (this study) under UV-light can achieve a high rate of 51.23 μmol g$^{-1}$ h$^{-1}$ (Supplementary Figure 25). To the best of our knowledge, the $CH_4$ evolution rate over the HCP-$TiO_2$-FG under UV- or visible-light irradiation is impressively higher than the reported values including that over precious metal-modified $TiO_2$ photocatalysts (Supplementary Table 1), under similar gas–solid reaction conditions (with no sacrificial agent).

The charge separation efficiency was investigated by recording the transient amperometric I-t curves under visible-light irradiation. As shown in Fig. 4e, the photocurrent of the resulting materials displayed high repeatability during light on-off cycling, and the results were consistent with the photocatalytic evaluations, i.e., negligible signal in pure $TiO_2$ system, weak photocurrent response in HCP-FG, and notably enhanced current intensity in HCP-$TiO_2$-FG composite. The photoluminescence (PL) and electrochemical impedance spectra (EIS) were employed to provide further evidence as shown in Supplementary Figures 26–27. The significant PL quenching suggests that the recombination of the photogenerated e$^-$/h$^+$ pairs was effectively suppressed by graphene. The smaller semicircle arc at high frequencies in the EIS indicates the decreased electron-transfer resistance ($R_{et}$) across the electrode/electrolyte. The lower $R_{et}$ of $TiO_2$-FG than that of $TiO_2$ indicates that FG modification favors the electronic conductivity due to its high electron mobility. Moreover, the covalent linking with graphene effectively improves the electronic conductivity of the HCPs and thus facilitates the electron transfer in the composite. The less efficient $CH_4$ production over HCP-$TiO_2$ photocatalyst can also reflect the

influence of graphene on improving the charge separation efficiency (Supplementary Figure 28). As a result, the porous HCP-TiO$_2$-FG composite possesses the improved efficiency in separating the photogenerated charge carriers.

The pathway of charge carriers transfer and separation generally depends on the band gap of photocatalysts. The HCP-FG showed that its highest occupied molecular orbital (HOMO) and lowest unoccupied molecular orbital (LUMO) energy levels were located at −5.34 eV and −3.00 eV (vs. vacuum level) as calculated by optical absorption (Fig. 4d) and cyclic voltammetry (CV) measurement (Supplementary Figure 29), which are more negative than the valence band (VB) and conduction band (CB) levels of TiO$_2$, respectively. To further confirm, ultraviolet photoelectron spectroscopy (UPS) technique was employed to measure the HOMO location, −5.44 eV vs vacuum level (Supplementary Figure 30), which was found to be very close to that of CV measurement. Based on the position of HOMO and LUMO energy levels, a tentative mechanism for the overall CO$_2$ conversion process over the HCP-TiO$_2$-FG photocatalyst is proposed and is shown in Fig. 4f. Under visible-light irradiation, HCP-FG functions both as CO$_2$ adsorbent and photosensitizer, which directly absorbs the photons to induce the HOMO to LUMO transition. The photogenerated electron-hole pairs of the excited HCP-FG can migrate and separated at the interface with TiO$_2$ via their interfacial interaction, as shown in Supplementary Figure 31. Thus the CO$_2$ reduction is inclined to occurr at the catalytic sites on TiO$_2$ rather than that on HCP-FG, which is in consistent with the above discussions on the porous property and photocatalytic performance. The electron transfer not only largely inhibited the recombination with the excited HCP-FG, but also made the photocatalytic reaction more effective. The excited HCP-FG was recovered to its neutral state by oxidizing the absorbed water molecules to produce oxygen gas. When the light was turned off, the amount of CO and CH$_4$ remained almost constant, indicating the efficient control over the reverse reactions of CO$_2$ reduction (oxidizing CO and CH$_4$ to CO$_2$) over the HCP-TiO$_2$-FG photocatalyst under such mild reaction conditions (Supplementary Figure 32). It can be concluded that the in situ knitting of porous HCP-TiO$_2$-FG are very effective to dramatically enhance visible-light-driven CO$_2$ conversion efficiency, which could be mainly ascribed to the well-defined composite structure from the following two aspects: (1) high CO$_2$ uptake ability and short distance between the adsorbent and the catalyst facilitating the CO$_2$ adsorption and diffusion; (2) broad light absorption of HCPs layers and fast charge mobility of graphene improving the visible-light absorption and charge separation efficiency. In this way, the high selectivity of electron consumption for CH$_4$ production can also be explained by the enrichment of CO$_2$ and electron density favoring the multi-electron reduction process.

In addition to the enhanced photocatalytic activity, the stability of such photocatalysts is also crucial for composite structure because of the possible leakage of one component from another. It was observed that HCP-TiO$_2$-FG retained more than 85% of the original efficiency for both the CH$_4$ and CO production, implying their stable framework by strong interfacial interaction after five consecutive runs (Supplementary Figure 33). Moreover, the fresh and used catalysts displayed no obvious difference in FT-IR spectra and XRD diffraction pattern, which demonstrates the stable chemical skeleton and crystal structure of HCP-TiO$_2$-FG throughout the photocatalytic reaction (Supplementary Figures 34–35).

**Kinetic analysis**. The kinetics experiments were carried out to understand the contribution of CO$_2$ adsorption and diffusion to

the enhancement of photocatalytic efficiency. The relationship between the CO$_2$ adsorption and CH$_4$ production can be explored by varying the surface coverage of CO$_2$ on the active sites. The partial pressure of CO$_2$ is adjusted in CO$_2$/N$_2$ mixture because of a high CO$_2$/N$_2$ selectivity ratio of 25.8 over the HCP-TiO$_2$-FG photocatalyst (Supplementary Figure 36). Since the kinetic model and reaction mechanism of photocatalytic CO$_2$ conversion are ambiguous so far, the quantitative relationship between CO$_2$ coverage and CH$_4$ evolution rate is still unclear. Interestingly, it is observed that they show a similar trend of increase with CO$_2$ proportion, e.g., both of them dramatically increased at lower partial pressure and then displayed a slow increase at higher CO$_2$ concentration (Supplementary Figure 37). Generally, the reaction rates that are normalized to the active sites allow the direct comparison of intrinsic reactivity on different catalysts[47–49]. For the catalytic system employing same catalyst, the reaction rate appears to be independent of the loading amount of catalyst after normalization to the same amount[50,51]. In this regard, the porous HCP-TiO$_2$-FG photocatalyst possesses equivalent catalytic active sites to TiO$_2$/HCP-FG due to the same content of TiO$_2$ photocatalyst. That is, the more efficient CH$_4$ production over HCP-TiO$_2$-FG should not result from the difference in the number of catalytic sites but mostly come from the higher surface coverage of CO$_2$ on the active sites.

The temperature has a complicated influence on the rate of photocatalytic conversion from the aspects of adsorption and diffusion. By increasing the temperature, the surface coverage of CO$_2$ molecules on the catalyst surface was decreased due to the exothermic effect of adsorption process (Fig. 3e–f), while the diffusion rate was increased as a result of the increased thermal motion of CO$_2$ molecules (Supplementary Figure 38). Based on Arrhenius plot, the adsorption activation energy for CO$_2$ adsorption is calculated to be 5.20 kJ mol$^{-1}$ (Supplementary Figure 39a) using a microporous diffusion model[52,53]. Since the CH$_4$ production increases linearly and possesses dominant electron consumption selectivity as 83.7%, we can use the pseudo-zero-order model to estimate the rate constant for the overall reaction, obtaining apparent activation energy of 9.34 kJ mol$^{-1}$ (Supplementary Figure 39b). The diffusion process was further studied by varying the stirring speed. As shown in Supplementary Figure 40, the increase of stirring speed greatly facilitates the photocatalytic conversion of CO$_2$ to CH$_4$ product. Combining the diffusion effect with pressure-/temperature-dependent characteristics, we can conclude that the photocatalytic CO$_2$ reduction over HCP-TiO$_2$-FG is not under intrinsic kinetic control of the catalyst, but the efficiency is rather determined by gas adsorption and diffusion. The elucidation of adsorption and diffusion that contributed to the photocatalytic reaction, and is seldom discussed in the literature, provides valuable information for understanding the relationship between the catalytic performance and structure properties. As a result, it clearly demonstrates the superiority of such porous HCP-TiO$_2$-FG composite towards the visible-light-driven photocatalytic CO$_2$ conversion. Further kinetic study is required to probe the kinetics model and reaction mechanism of photocatalytic CO$_2$ conversion.

In summary, the well-defined porous HCP-TiO$_2$-FG composite structure was successfully constructed by in situ knitting strategy. The anatase TiO$_2$ crystals with reactive {001} facets were supported on graphene surface and encapsulated inside the ultrathin HCPs layers with a thickness of 3–8 nm. Benefiting from the high surface area and abundant microporous nature, the introduction of HCPs layers dramatically improved the specific surface area and micropore volume of TiO$_2$-G to 988 m$^2$ g$^{-1}$ and 0.306 cm$^3$ g$^{-1}$, respectively, leading to the increased CO$_2$ uptake capacity up to 12.87 wt%. Due to the improved CO$_2$ adsorption

ability and shortened diffusion length, such well-defined HCP-$TiO_2$-FG composite photocatalyst is expected to enhance the reactivity of $CO_2$ molecules, which will facilitate the $CO_2$ conversion especially for $CH_4$ production. In addition, the charge separation efficiency and visible-light absorption of $TiO_2$ photocatalyst could be effectively improved by the graphene with high charge mobility and the HCPs layers with broad light absorption. As a result, the HCP-$TiO_2$-FG achieved high $CO_2$ conversion efficiency with a rate of total consumed electron number ($R_e$) as 264 $\mu mol\ g^{-1}\ h^{-1}$, including 83.7% selectivity for $CH_4$ production and negligible side reaction of $H_2$ production under visible-light irradiation. To the best of our knowledge, these results are the best among the recent such reports, and especially much better than those of photocatalysts with precious metal co-catalysts under similar gas–solid reaction conditions. We believe that these findings will be very helpful to overcome the constraint of deficient pore structure for semiconductor-based composites and open a new pathway for the design and synthesis of well-defined porous materials with high $CO_2$ uptake and photocatalytic conversion efficiency.

## Methods

**Materials**. Graphite, isopropyl alcohol, dichloromethane, hydrochloric acid, fluoric acid, ethanol, $AlCl_3$ (anhydrous), $H_2SO_4$, $NaNO_3$, $KMnO_4$, and $K_2CO_3$ were obtained from National Medicines Corporation Ltd. of China, all of which were of analytical grade and were used as received. 1, 3, 5-triphenylbenzene (*syn*-PhPh$_3$), tetrabutyl titanate, isoamyl nitrite, and aniline were purchased from Aladdin chemical reagent Corp (Shanghai, China) and used as received.

**Synthesis of graphene supporting $TiO_2$ with reactive {001} facets ($TiO_2$-G)**[25,26]. The graphene oxide (GO) was synthesized via a modified hummer's method (Supplementary Methods)[54]. The lamellar protonated titanate (LPT) was used as the precursor of $TiO_2$ (Supplementary Methods). 2.5 g of wet LPT precursor and 30 mg GO (6 mg/mL) were dissolved in 15 mL of isopropyl alcohol by sonication for 30 min. Then, 0.5 mL of fluoric acid (40 wt%) and 16 mg of glucose were dropped into the solution under continuous stirring. The mixed solution was subjected to solvothermal treatment at 180 °C for 12 h. When the reaction system was cooled to room temperature, the black precipitate, $TiO_2$-G, was washed thoroughly with water and ethanol, and then dispersed in absolute ethanol for the next step.

**Creation of porous composite structure by in situ knitting hypercrosslinked polymers on $TiO_2$-functionalized graphene (HCP-$TiO_2$-FG)**. Firstly, the phenyl-substituted graphene supporting $TiO_2$ ($TiO_2$-FG) was obtained by the functionalization of graphene skeleton[55]. 5 millimoles of aniline (2 equivalent per graphene carbon), 100 mg $TiO_2$-G, and 30 mL acetonitrile were added to a 50 mL round-bottom flask and stirred for 60 min. 5 millimoles of isoamyl nitrite was added under nitrogen atmosphere, and then the reaction mixture was heated with stirring to 80 °C for 24 h. The $TiO_2$-FG was obtained by filtration and washing thrice with chloroform, further purified by extracting with chloroform for 24 h, and finally dried in a vacuum oven at 60 °C for 24 h. Secondly, the knitting process was performed as follows: under a $N_2$ atmosphere, the catalyst ($AlCl_3$, 104 mg, 12 equiv *syn*-PhPh$_3$) was added to a mixture of *syn*-PhPh$_3$ (20 mg) and $TiO_2$-FG (40 mg) in dichloromethane (8 mL), and then the system connected to nitrogen-line to form relatively less air-controlled environment. The reaction system was then stirred at 0 °C for 4 h, 30 °C for 8 h, 40 °C for 12 h, 60 °C for 12 h, and 80 °C for 24 h to obtain the resulting material. The resulting precipitate was quenched using HCl-$H_2O$ ($V/V = 2:1$), and washed thrice with water and twice with ethanol, extracted with ethanol for 48 h, and finally dried in a vacuum oven at 65 °C for 24 h. The amount of *syn*-PhPh$_3$ was changed to 25 and 30 mg to adjust the thickness of HCP outer layers in the HCP-$TiO_2$-FG composite structure. The resulting samples were labeled as HCP-$TiO_2$-FG-1 and HCP-$TiO_2$-FG-2, respectively.

**Synthesis of $TiO_2$/HCP-FG composite**. For comparison, another type of composite, $TiO_2$/HCP-FG, was prepared by a common synthetic method to illustrate the superiority of such well-defined HCP-$TiO_2$-FG composite structure for $CO_2$ uptake. That is, *syn*-PhPh$_3$ was knitted on FG and then the HCP-FG was used as a supporting material for $TiO_2$ crystals growth. Firstly, hydrazine hydrate was used to reduce GO to graphene. GO (225 mg) was dispersed in 1 wt% aqueous sodium dodecylbenzene-sulfonate surfactant solution (225 mL) and sonicated for 60 min. The GO dispersion was reduced with 98% hydrazine hydrate (4.5 mmol) at 100 °C for 24 h after its pH was adjusted to 10 by 5% NaOH solution. Secondly, the HCP-FG was synthesized by the above diazonium salt process and solvent knitting method (20 mg FG and 20 mg *syn*-PhPh$_3$). Finally, the $TiO_2$/HCP-FG composite

was obtained by the same solvothermal process as $TiO_2$-G by adding HCP-FG instead of GO.

**Photocatalytic test for $CO_2$ conversion**. Photocatalytic activity for $CO_2$ conversion of the synthesized catalysts was evaluated in a closed gas reactor. A 300 W Xe lamp was used as the light source. For visible light photocatalysis, a cutoff filter was used to remove any radiation below 420 nm. Before the photocatalytic reaction, the catalyst powder (20 mg) was placed in a circular glass dish that was positioned 8 cm away from the light source. The optical density was measured to be 433 mW cm$^{-2}$ and the illuminated area of photocatalyst was about 3.14 cm$^2$. The $CO_2$ was generated by the reaction of sodium hydrogen carbonate with diluted sulphuric acid after removing the air. The gas evolutions were analysed by gas chromatography (GC-2014CA, Shimadzu Corp., Japan).

**Characterization of materials**. FT-IR spectra were recorded on a Bruker Vertex 70 Spectrometer employing the KBr disk method. Thermogravimetric analysis (TGA) was performed from room temperature to 850 °C under nitrogen and air, employing a Perkin Elmer Instrument Pyris1 TGA with a heating rate of 10 °C min$^{-1}$. The field-emission scanning electron microscopy (FE-SEM) images were recorded employing a FEI Sirion 200 field-emission scanning electron microscope operated at 10 kV. The X-ray powder diffraction (XRD) (PANalytical B.V.) spectra were recorded by using an x'pert3 powder equipped with Cu Kα radiation at a scanning rate of 4° min$^{-1}$. X-ray photoelectron spectroscopy (XPS) measurements were performed on AXIS-ULTRA DLD-600W with an Al Kα source. The high-resolution transmission electron microscopy (HR-TEM) and scanning transmission electron microscopy (STEM) images of samples were recorded on a Tecnai G2 F30 microscope (FEI Corp. Holland). The High-Angle Annular Dark Field (HAADF) mapping images and three-dimensional TEM (3D-TEM) movie were obtained on a Talos F200X field-emission transmission electron microscope. The 3D-TEM movie in the Supplementary Movie 1 was created by taking multiple views of the sample at differ angles from −50° to 50° with 2° increments. The actual loading amount of Ti was measured by inductively coupled plasma-mass spectrometry (NexION 300X, Perkin Elmer). UV-vis diffuse reflectance spectra (DRS) were obtained on a scanning UV-vis spectrometer (LabRAM HR800) with integrating sphere detector. Gas ($H_2$, $N_2$, $CO_2$) sorption properties and specific surface area of samples were measured using a Micromeritics ASAP 2020 surface area and porosity analyzer. Before analysis, the samples were degassed at 110 °C for 8 h under vacuum of 10$^{-5}$ bar. Pore size distribution was calculated by $N_2$ adsorption isotherm employing the Tarazona nonlocal density functional theory (NLDFT) model assuming slit pore geometry. Total pore volumes ($V_{toal}$) were derived from nitrogen sorption isotherms at relative pressure $P/P_0 = 0.995$. The mix gas adsorption experiment were checked by volumetric adsorption-gas chromatograph instrument BELSORP-VC. The samples were degassed at 100 °C for 8 h to remove the remnant solvent molecules before measurement. The pure water vapor adsorption experiment was investigated by moisture sorption analyser Hiden IGAsorp. The isotopic labelling was confirmed using a gas chromatography-mass spectrometry (SHIMADZU GCMS-QP2020). The Ultraviolet photoelectron spectroscopy (UPS) measurement was performed on a VG Scienta R4000 analyzer using a monochromatic He I light source (21.2 eV). A sample bias of −5 V was applied to observe the secondary electron cutoff (SEC).

## Data availability

The data supporting the plots within this paper and other findings of this study are available from the corresponding authors on request. The source data underlying Figs. 3a–f and 4a–e and Supplementary Figs. 7–12, 14, 16, 18–23, 25–30 and 32–40 are provided as a Source Data file, which is available at figshare website (the unique identifier for the data is https://doi.org/10.6084/m9.figshare.7527764, and the permanent web address is https://figshare.com/s/03cf28e67f620dce6358).

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

## Acknowledgements

We thank the Analysis and Testing Center, Huazhong University of Science and Technology for characterization assistance in characterization of materials. We also thank Prof. Ming-Tian Zhang of Tsinghua University for the quantitative measurement of $O_2$ gas. This work was supported by the Program for National Natural Science Foundation of China (No. 21771070, 21571071, 21474033), the International S&T Cooperation Program of China (2016YFE0124400), the China Postdoctoral Science Foundation funded project (No. 2017M622423), the Fundamental Research Funds for the Central Universities (2018KFYYXJJ120), and the Program for HUST Inter-disciplinary Innovation Team (2016JCTD104).

## Author contributions

B.T. and J.W. conceived the project and designed the experiments. S.W. and M.X. performed the experiments and analyzed the data. C.Z. conducted part of synthesis. T.P. designed the photocatalytic measurements. T.L. and I.H. discussed the results and helped in improving the manuscript. S.W., M.X., J.W., and B.T. co-wrote the manuscript. S.W. and M.X. contributed equally to this work.

## Additional information

**Competing interests:** The authors declare no competing interests.

**Journal Peer Review Information**: *Nature Communications* thanks the anonymous reviewers for their contribution to the peer review of this work. Peer reviewer reports are available.

