## [Peer Review File · Nature Communications]

Reviewers' comments:

Reviewer #1 (Remarks to the Author):

This paper reports modified TiO₂-graphene as a visible-light-sensitive photocatalyst. Its idea is interesting, however, there are some cogent issues to be addressed and the present paper cannot be considered for publication in Nature Communications as follows,

1. The authors did not discuss oxidation reaction at all. Possible candidate of oxidation product is oxygen, but their photocatalyst could not oxidize water according to their band diagram shown in Figure 4. What is electron donor for CO₂ reduction in the authors' system? The authors should show the evidence of oxidation products.

2. The reviewer wonders if the origin of CO production might be polymer degradation even though they showed good repeated property. The authors described that CO evolution increased in the absence of water, which implies that CO generation would be induced by partial oxidation of polymer or graphene. The reviewer recommends the authors to conduct isotope labeling experiment using carbon thirteen CO₂ to show that the origin of their product would be CO₂.

3. The authors emphasized that their originality is the efficient adsorption of CO₂, thus, they should show the reasonable experimental evidence like FT-IR to claim their efficient CO₂ adsorption.

Reviewer #2 (Remarks to the Author):

Comments:

This work by Wang et al. presents the fabrication of a porous hypercrosslinked polymer-TiO₂-graphene (HCP-TiO₂-FG) sandwiched structure for visible-light-driven CO₂ conversion. The structure possesses relatively high surface area and CO₂ uptake capacity, and shows relatively high photocatalytic performance for CO and CH₄ production without the use of sacrificial reagents or precious metal co-catalysts. The authors argue that the superb catalytic activity is attributed to the improved CO₂ adsorption and diffusion, visible-light absorption and photo-generated charge separation. Indeed, it is a topic of interest to the researchers in the related areas; however, the manuscript still cannot meet the level of Nature Communications. No deep understanding and important scientific issues have been given in this manuscript. Several issues need to be clarified, and part of the conclusions cannot be supported by the current data.

Specific comments are listed below:

1. The authors have stated that this is the first example involving microporous organic polymers for CO₂ conversion among numerous photocatalysts. This may overstate the significance of this work. As far as I know, conjugated microporous polymers have been used as materials for the capture and conversion of CO₂ (Nature Communications, DOI: 10.1038/ncomms2960) and visible-light-driven conversion of CO₂ (Green Chem., 2017, 19, 5777).

2. The authors have demonstrated that TiO₂ crystals are supported on graphene sheets and encapsulated by ultrathin HCPs layer. As such, the TiO₂ should be in the middle of the sandwiched structure. As stated in the manuscript, the photogenerated electrons migrate from HCP to TiO₂ via their interfacial interaction. In this case, what is the role of the graphene? How can graphene improve the charge separation efficiency? The authors should clarify these issues in the manuscript.

3. In “Characterization of the resulting materials” part, the authors have stated that the percentage of the exposed {001} facets in the TiO₂ crystal is calculated to be approximately 30%. What is the relationship between the {001} facets and the photocatalytic performance for CO₂ conversion? If the {001} facets are more reactive, would it be better to use TiO₂ nanosheets with dominant (001) facets?

4. In thermogravimetric analysis (TGA, Figure S9), the authors have stated that the HCP-TiO₂-FG composite structure exhibits excellent thermal stability compared to TiO₂-G with resistance of degradation up to 400 °C. However, according to Figure S9, the thermal stability of TiO₂-G is even better than HCP-TiO₂-FG with resistance of degradation up to 400 °C.

5. For the photocatalytic test, the authors should provide the details for the optical density of light source and the illuminated area. In addition, to verify the origins of produced CO and CH₄, the isotopic ¹³CO₂ must be used as a reactant to trace the carbon sources in the photocatalytic reaction.

6. According to Figure 4a and b, the HCP-FG material also exhibits a photocatalytic activity for CO and CH₄ production. The authors have stated that the catalytic sites are located on TiO₂. In this case, it remains elusive whether the catalytic sites are located on TiO₂ or HCP-FG. The authors should clarify this point.

7. In Figure 4f, the authors have used the band gap energy, HOMO and LUMO levels of HCP-FG to discussing the charge separation and transfer within the HCP-TiO₂-FG sandwich photocatalyst. This analysis is misleading. According to the sandwiched structure of HCP-TiO₂-FG, the photogenerated electrons migrate from HCP to TiO₂ via their interfacial interaction. As a result, the authors should use the band gap energy, HOMO and LUMO levels of HCP rather than HCP-FG in the analysis.

8. According to the time-dependent production of CH₄ and CO in Figure S13, it remains unclear why the production of CH₄ increases linearly while the production of CO has a stagnation effect.

9. In the electrochemical impedance spectra (EIS, Figure S15), HCP-FG has a much larger semi-circle diameter than HCP-TiO₂-FG. Why the addition of TiO₂ can dramatically reduce the electron-transfer resistance?

10. In abstract, “...and highlights the importance of MOPs in combination with photocatalysts for solar energy conversion.”, there is no definition for the “MOPs” in the manuscript.

Reviewer #3 (Remarks to the Author):

The authors report on a novel catalyst for photocatalytic CO₂ activation using visible light enabling unprecedented activity and CH₄ selectivity. This high performance is achieved by tailoring the CO₂ adsorption of the material together with short diffusion length to the active sites.

To achieve these features, the authors prepare a composite material composed of TiO₂ anatase crystals with exposed {001} facets as active sites grown on graphene oxide. The latter was functionalized by PhPh₃ serving as anchoring groups enabling coating of a thin layer of hypercrosslinked polymer (HCP).

The material is comprehensively characterized. Its catalytic activity is reported compared to pure

titania as well as further references such as the composite without titania, titania supported on graphene oxide-HCP composite etc. The results are presented in a systematic and sound manner.

Concerning the impact, the contribution is clearly important but justification for a publication in Nature Communications is not sufficiently provided. In addition, the hypothesis of the contribution is based on CO₂ adsorption and diffusion length as crucial parameters to enhance catalytic activity. Following this line of argument, the major aspects to critically consider are:

- CO₂ adsorption on or close to the active site is important, as surface coverage presents the concentration subsequently determining the reaction rate to be achieved ($r = k \cdot \text{surface coverage of reagents}$, considering a Langmuir Hinshelwood type of activation). In line, a higher surface coverage of the substrates enables reaching a higher reaction rate, while the intrinsic catalytic activity of the active site remains unaltered. Following this argument, the authors have to carefully proof their hypothesis that CO₂ adsorption within the material indeed causes enhanced coverage on or close to the active sites, increasing reaction rate. Kinetic experiments varying partial pressure, temperature etc. and providing assessment of the influence on the reaction rate are indispensable to justify the hypothesis. In addition, a careful representation of the literature state of the art including rates achieved in previous contributions normalized to the active sites content of the different catalysts are essential.

- The second argument relates to diffusion length as important element of catalyst design. The provided data do not allow any conclusion of the role of surface diffusion within the overall system. Following the argument of a diffusion governed process, the rate of surface diffusion should be somewhat rate determining. Consequently, kinetic analysis following e.g. the temperature dependence are needed to support such a hypothesis. Diffusion limitation should cause a limited apparent activation energy according to the reduced temperature dependence of various types of diffusion compared to chemical reactions.

- As a minor point, the experimental data are not provided with sufficient information to understand the used experimental setup, e.g. in the related figure captions. Where the reactions carried out in a flow reactor or a batch system? What was the temperature? Partial pressure of CO₂ and H₂O were constant (was there water in the system)?

- Concerning CO₂ adsorption experiments, data in presence of water appear important and water vapor sorption experiments would be complementary.

Response to Reviewers' Comments

Below are our responses (in BLUE colour) to the Reviewers' comments.

Reviewer 1:

This paper reports modified TiO₂-graphene as a visible-light-sensitive photocatalyst. Its idea is interesting, however, there are some cogent issues to be addressed and the present paper cannot be considered for publication in Nature Communications as follows,

Response: We thank the reviewer for appreciating the idea of work presented in this manuscript. We also appreciate the reviewers' comments and suggestions that helped us to significantly improve the manuscript. We have addressed the reviewers' comments point-wise in our response below:

1. The authors did not discuss oxidation reaction at all. Possible candidate of oxidation product is oxygen, but their photocatalyst could not oxidize water according to their band diagram shown in Figure 4. What is electron donor for CO₂ reduction in the authors' system? The authors should show the evidence of oxidation products.

Response: This suggestion encouraged us to further investigate the oxidation reaction and explore the possible mechanism in the photocatalytic CO₂ conversion. As shown in supplementary information (**Supplementary Fig. S28-29**), the excellent stability of porous HCP-TiO₂-FG and its resistance to oxidative degradation indicate that the excited HCP-FG molecules are mostly recovered to its neutral state by accepting the electrons from water, and thus the possible electron donors are water molecules and the oxidation product should be oxygen. The band diagram of HCP-FG was examined to find out whether it could drive the water oxidation reaction. We realized that the dissolved O₂ and trace water in the previous CV measurement would cause some deviations in energy levels. This time the CV measurements were performed carefully in N₂-saturated anhydrous CH₂Cl₂ solvent, thus the influence of the dissolved O₂ and trace water could be excluded. The HCP-FG shows that its highest occupied molecular orbital (HOMO) is located at -5.34 eV (vs. vacuum level) (**Supplementary Fig. S25**). Ultraviolet photoelectron spectroscopy (UPS) technique was employed to measure the energy of HOMO i.e., -5.44 eV vs vacuum level (**Supplementary Fig. S26**) that is very similar to that obtained from CV measurement. Consequently, the potential of the

HOMO level is found to be enough for water oxidation.

The quantitative measurement of O₂ evolution was conducted using an optical fiber oxygen sensor. The O₂ evolution rate over HCP-TiO₂-FG under visible-light irradiation was determined to be 1.6 μmol h⁻¹, while the O₂ evolution over other photocatalysts was too low to be detectable (**Supplementary Fig. S13**). The electrons from the water oxidation are slightly higher than the total consumed electrons for the reduced products including CH₄ and CO. To the best of our knowledge, this is the first example achieving the quantitative detection of the oxygen production during photocatalytic CO₂ conversion in such a gas-solid reaction system.

Moreover, the isotopic labeled H₂¹⁸O vapors was used to verify the origin of the detected O₂. The formation of ¹⁸O₂ suggests that the evolved O₂ gas is derived from the photocatalytic water oxidation (**Supplementary Fig. S16**). Finally, we corrected the energy levels and proposed a mechanism for the overall CO₂ conversion process over the HCP-TiO₂-FG photocatalyst as shown in **Fig. 4f**.

In the revised manuscript, we have added the following text on page 9 and 11:

*“As a result of the relatively high photocatalytic performance of porous HCP-TiO₂-FG, the O₂ evolution can be measured to provide the evidence of the oxidation cycle offering a better insight of the mechanism that is seldom discussed in the literature⁴³. The O₂ evolution rate over HCP-TiO₂-FG under visible-light irradiation was determined to be 1.6 μmol h⁻¹, while the O₂ evolution over other photocatalysts was too low to be detectable (**Supplementary Fig. S13**). The electrons from the water oxidation are slightly higher than the total consumed electrons for the reduced products including CH₄ and CO.”*

*“In an isotopically labeled experiment, the ¹³CH₄ and ¹³CO signals at m/z = 17 and m/z = 29 appeared after the photocatalytic reaction. The results confirmed that the CO and CH₄ products are indeed originating from the photocatalytic reduction of CO₂ gas (**Supplementary Fig. S15**). The isotopic labeled H₂¹⁸O vapors led to the formation of ¹⁸O₂ (**Supplementary Fig. S16**), suggesting that the evolved O₂ gas was derived from the photocatalytic water oxidation.”*

*“The HCP-FG showed that its highest occupied molecular orbital (HOMO) and lowest unoccupied molecular orbital (LUMO) energy levels were located at -5.34 eV and -3.00 eV (vs. vacuum level) as calculated by optical absorption (**Fig. 4d**) and cyclic voltammetry (CV) measurement (**Supplementary Fig. S25**), which are more negative than the valence band (VB) and conduction band (CB) levels of TiO₂ respectively. To*

further confirm, ultraviolet photoelectron spectroscopy (UPS) technique was employed to measure the HOMO location, -5.44 eV vs vacuum level (**Supplementary Fig. S26**), which was found to be very close to that of CV measurement. Based on the position of HOMO and LUMO energy levels, a tentative mechanism for the overall CO_2 conversion process over the HCP- TiO_2 -FG photocatalyst is proposed and is shown in **Fig. 4f**.”

“The excited HCP-FG was recovered to its neutral state by oxidizing the absorbed water molecules to produce oxygen gas.”

Figure S13. The O_2 evolution rate during photocatalytic CO_2 conversion over photocatalysts under visible-light irradiation ($\lambda \geq 420\text{ nm}$). The quantitative measurement of O_2 gas was conducted using an optic fiber oxygen sensor (Ocean-Optics).

Figure S16. GC-MS spectra of $^{18}\text{O}_2$ ($m/z=36$) after the photocatalytic reaction over HCP- TiO_2 -FG under visible-light irradiation ($\lambda \geq 420\text{ nm}$). The isotopic labeled H_2^{18}O was used as the substrate in normal experiment condition.

Figure S25. Cyclic voltammetry (CV) measurements of ferrocene (FC) reference and HCP-TiO₂-FG catalyst on ITO glass electrode as the work electrode. The CV curves were recorded in N₂-saturated anhydrous dichloromethane (CH₂Cl₂) containing 0.1 M tetrabutyl-ammonium hexafluorophosphate (Bu₄NPF₆). The ITO glass electrode with no catalyst coating was labeled as blank for comparison. Reference electrode: Ag/AgCl. Counter electrode: Pt wire. Scan rate: 100 mV/s.

The energy gap of HCP-TiO₂-FG can be determined from the UV-vis spectrum in **Fig. 4d** using the equation $E_g = 1240/\lambda = 1240/530 = 2.34$ eV. The potentials levels vs. the vacuum level can be obtained from the above CV curves using the following equations:

$$\text{HOMO} = -[E_{\text{ox}} - E(\text{FC}/\text{FC}^+) + 4.8] \text{ eV} = -(0.93 - 0.39 + 4.8) \text{ eV} = -5.34 \text{ eV}$$

$$\text{LUMO} = -(\text{HOMO} + E_g) = -(-5.34 + 2.34) = -3.00 \text{ eV}$$

Figure S26. Ultraviolet photoelectron spectroscopy (UPS) measurement. (a) secondary electron cutoff spectrum and (b) highest occupied molecular orbital (HOMO) energy level.

The work function (Φ) can be determined by the difference between the photon energy (21.2 eV) and the binding energy of the secondary cutoff edge.

$$\Phi = 21.20 - 16.86 = 4.34 \text{ eV}$$

The HOMO location is measured to be 1.10 eV below the Fermi level (E_F), corresponding to -5.44 eV vs vacuum level.

Figure 4(f). Proposed mechanism of charge separation and transfer within the HCP-TiO₂-FG sandwich photocatalyst under visible-light ($\lambda \geq 420$ nm) irradiation.

2. The reviewer wonders if the origin of CO production might be polymer degradation even though they showed good repeated property. The authors described that CO evolution increased in the absence of water, which implies that CO generation would be induced by partial oxidation of polymer or graphene. The reviewer recommends the authors to conduct isotope labeling experiment using carbon thirteen CO₂ to show that the origin of their product would be CO₂.

Response: This is a good suggestion. As suggested, we have conducted the isotope labeling experiment by using ¹³CO₂ as the substrate to detect the origin of the products. The results indicate that the origin of CO and CH₄ production is CO₂ instead of polymer degradation (**Supplementary Fig. S15**).

We have now also included the corresponding description on page 9 in the manuscript.

“To verify the evolution of CO and CH₄ from CO₂ conversion over HCP-TiO₂-FG photocatalyst, we conducted three controlled experiments: (1) irradiation of catalyst under inert N₂ condition; (2) the use of isotopic labeled ¹³CO₂ and H₂¹⁸O as the reactants; (3) irradiation of catalyst in the presence of CO₂ gas without H₂O vapors.”

*“In an isotopically labeled experiment, the ¹³CH₄ and ¹³CO signals at m/z = 17 and m/z = 29 appeared after the photocatalytic reaction. The results confirmed that the CO and CH₄ products are indeed originating from the photocatalytic reduction of CO₂ gas (**Supplementary Fig. S15**).”*

Figure S15. GC-MS spectra of $^{13}\text{CH}_4$ ($m/z=17$) and ^{13}CO ($m/z=29$) after the photocatalytic reaction over HCP-TiO₂-FG under visible-light irradiation ($\lambda \geq 420$ nm). The isotopic labeled $^{13}\text{CO}_2$ was used as the substrate in normal experiment condition.

3. The authors emphasized that their originality is the efficient adsorption of CO₂, thus, they should show the reasonable experimental evidence like FT-IR to claim their efficient CO₂ adsorption.

Response: In most of the cases, FT-IR analysis was done to study the chemisorption of CO₂ molecules on the catalysts. It is well established that the physical adsorption property of porous polymer materials can be evaluated by CO₂ adsorption/desorption isotherms. The CO₂ adsorption/desorption isotherms in **Fig. 3e-f** indicate the efficient adsorption of CO₂ on HCP-TiO₂-FG as 12.87 wt% at 1.00 bar and 273.15 K, which is more than 4-fold higher than that of TiO₂ and TiO₂-G.

Moreover, we have also conducted TGA-DSC analysis under CO₂ atmosphere to get further evidence for CO₂ adsorption. The results are shown in **Supplementary Fig. S11**. The adsorption mode displays the pressure and temperature-dependent features with excellent recyclability for the repeated CO₂ adsorption and desorption. Based on the literature reports (*Energy Environ. Sci.* 2014, 7, 3478; *Chem. Soc. Rev.* 2017, 46, 3322), the CO₂ adsorption by such porous polymer materials is in consistent with the characteristics of physical adsorption, which cannot affect the signals in FT-IR spectra. We have now also included corresponding description on page 6 in the manuscript:

*“It is well established that the CO₂ uptake by porous polymer materials mainly results from its physical adsorption^{34, 35}. Such adsorption mode displays the pressure and temperature-dependent features with excellent recyclability for the repeated CO₂ adsorption and desorption (**Supplementary Fig. S11**).”*

Figure S11. The temperature-dependent adsorption ability of porous HCP-TiO₂-FG checked by TGA-DSC at CO₂ atmosphere.

Reviewer 2

This work by Wang et al. presents the fabrication of a porous hypercrosslinked polymer-TiO₂-graphene (HCP-TiO₂-FG) sandwiched structure for visible-light-driven CO₂ conversion. The structure possesses relatively high surface area and CO₂ uptake capacity, and shows relatively high photocatalytic performance for CO and CH₄ production without the use of sacrificial reagents or precious metal co-catalysts. The authors argue that the superb catalytic activity is attributed to the improved CO₂ adsorption and diffusion, visible-light absorption and photo-generated charge separation. Indeed, it is a topic of interest to the researchers in the related areas; however, the manuscript still cannot meet the level of Nature Communications. No deep understanding and important scientific issues have been given in this manuscript. Several issues need to be clarified, and part of the conclusions cannot be supported by the current data.

Response: Thanks much for appreciating the quality of work presented in this manuscript. We believe that this work is novel and would have great impact in the field. To date, there is no report on the combination of microporous organic polymers with photocatalytic materials for CO₂ uptake and conversion. The photocatalytic performance is dramatically enhanced by the design and synthesis of porous HCP-TiO₂-FG sandwiched photocatalysts. Based on the results obtained, we have presented

the understanding of such superior structure from the aspects of improving CO₂ adsorption and diffusion, visible-light absorption, and photogenerated charge separation efficiency. These factors are widely studied scientific issues in the related areas. In the revised version, we have conducted the kinetic experiments to provide further understanding. We also thank the reviewer's kind suggestions. We have addressed all the issues point-by-point in our response below.

Specific comments are listed below:

1. The authors have stated that this is the first example involving microporous organic polymers for CO₂ conversion among numerous photocatalysts. This may overstate the significance of this work. As far as I know, conjugated microporous polymers have been used as materials for the capture and conversion of CO₂ (*Nature Communications*, DOI: 10.1038/ncomms2960) and visible-light-driven conversion of CO₂ (*Green Chem.*, 2017, 19, 5777).

Response: While we appreciate the reviewer's comments, we want to emphasize that the present work is fundamentally different from those reports i.e., *Nature Commun.*, 2013, DOI: 10.1038/ncomms2960 and *Green Chem.*, 2017, 19, 5777.

The microporous polymers in the above-mentioned *Nature Commun* are developed as heterogeneous catalysts for the reaction of CO₂ and propylene oxide for the formation of propylene carbonate. Indeed, there are many such reports on the use of microporous polymeric catalysts for chemical conversion of CO₂ (*Chem. Commun.* 2015, **51**, 11576; *Adv. Mater.* 2017, **29**, 1700445; *J. Mater. Chem. A* 2018, **6**, 374; *et al*). We also reported a metalporphyrin-based microporous polymer for catalyzing the reaction of CO₂ with propylene oxide (*J. Mater. Chem. A* 2017, **5**, 1509). Compared to the chemical conversion, the photocatalytic CO₂ reduction is of great significance because it utilizes the abundant and sustainable solar energy to produce carbonaceous fuels.

Another report in the above-mentioned "*Green Chem*" journal is related to the photocatalytic conversion of CO₂. However, the optimized pyrene-based polymer catalyst do not display porosity because it has extremely low surface area i.e., 23.9 m² g⁻¹. The photoreduction of CO₂ depends on the chemical capture of CO₂ molecules by a task-specific ionic liquid i.e., [P₄₄₄₄][p-2-O]. In addition, CO and H₂ are detected as the main products from the photocatalytic system, whereas the desired CH₄ production is absent. Thus the idea of using microporous organic polymers for CO₂ uptake and

photocatalytic conversion is missing in this work.

The previous studies were indeed focusing more on developing polymer materials for catalyzing CO₂ conversion. These findings inspired us to incorporate the microporous organic polymers as CO₂ capture materials into the photocatalytic system. As a result, we achieved high CO₂ conversion efficiency with a rate of total consumed electron number (R_e) as 264 $\mu\text{mol g}^{-1} \text{h}^{-1}$, including 83.7% selectivity for CH₄ production and negligible side reaction of H₂ production under visible-light irradiation.

To date, there is no report on the combination of microporous organic polymers with photocatalytic materials for CO₂ uptake and conversion. We believe this strategy will be very helpful to overcome the constraint of deficient pore structure for semiconductor-based composites and to open a new pathway for the design and synthesis of well-defined porous materials with high CO₂ uptake and photocatalytic conversion efficiency.

2. The authors have demonstrated that TiO₂ crystals are supported on graphene sheets and encapsulated by ultrathin HCPs layer. As such, the TiO₂ should be in the middle of the sandwiched structure. As stated in the manuscript, the photogenerated electrons migrate from HCP to TiO₂ via their interfacial interaction. In this case, what is the role of the graphene? How can graphene improve the charge separation efficiency? The authors should clarify these issues in the manuscript.

Response: The TiO₂ crystals are located in the middle of the sandwiched structure because they are supported on graphene nanosheets and encapsulated by ultrathin HCPs layer (**Fig. 2** and **Supplementary Fig. S1-4**). It should be pointed out that the HCPs and graphene are not freestanding in the composite. Benefiting from the *in-situ* knitting strategy, ultrathin HCPs layers were integrated with the functionalized graphene nanosheets through the methylene linkers. The structures have been verified by XPS, FT-IR, and CP/MAS NMR characterizations (**Fig. 3b-c** and **Supplementary Fig. S6-S8**).

As far as the role of the graphene is concerned, we have now added the EIS analysis of HCPs and HCPs-TiO₂, which are compared with HCPs-FG and HCPs-TiO₂-FG (**Supplementary Fig. S23**). The covalent linking with graphene effectively improves the electronic conductivity of the HCPs and thus facilitates the electron transfer in the composite. The comparison of photocatalytic performance between HCP-TiO₂ and HCP-TiO₂-FG photocatalysts is also shown in **Supplementary Fig. S24**. The less

efficient CH₄ production over HCP-TiO₂ photocatalyst can also reflect the influence of graphene on improving the charge separation efficiency.

The corresponding description has been revised on page 10 and 11 in the manuscript as:

“The lower R_{et} of TiO₂-FG than that of TiO₂ indicates that FG modification favors the electronic conductivity due to its high electron mobility. Moreover, the covalent linking with graphene effectively improves the electronic conductivity of the HCPs and thus facilitates the electron transfer in the composite. The less efficient CH₄ production over HCP-TiO₂ photocatalyst can also reflect the influence of graphene on improving the charge separation efficiency (Supplementary Fig. S24). As a result, the porous sandwich structure possesses the improved efficiency in separating the photogenerated charge carriers.”

Figure S23. Electrochemical impedance spectra (EIS) of the samples.

Figure S24. Comparison of CH₄ and CO production rates between HCP-TiO₂ and HCP-TiO₂-FG photocatalysts under visible-light ($\lambda \geq 420$ nm).

3. In “Characterization of the resulting materials” part, the authors have stated that the percentage of the exposed {001} facets in the TiO₂ crystal is calculated to be approximately 30%. What is the relationship between the {001} facets and the photocatalytic performance for CO₂ conversion? If the {001} facets are more reactive, would it be better to use TiO₂ nanosheets with dominant (001) facets?

Response: The exposed crystal facets have great impact on the photocatalytic performance of TiO₂ crystals. In the studies of crystal facet engineering of anatase TiO₂, both theoretical and experimental evidence demonstrates that the {001} facets are much more reactive but less thermodynamically stable than {101} facets due to higher average surface energy of the {001} facets than that of the {101} facets. As reported, the products with large percentage of {001} facets usually have a large size of micrometers or hundreds of nanometers and low surface area (*Nature* 2008, **453**, 638; *J. Am. Chem. Soc.* 2009, **131**, 4078; *J. Am. Chem. Soc.* 2009, **131**, 3152; *Chem. Commun.* 2009, **29**, 4381). When decreasing the particle size, the specific surface area increased, whereas the high-energy {001} facets tend to transform to the more thermodynamically stable {101} facets to reduce the high surface energy. For example, the TiO₂ crystals with size of ~20 nm only exposed 9.6% {001} facets, but the photoactivity could be comparable to that of micro-sized TiO₂ with dominant {001} facets (*Nano Lett.* 2009, **9**, 2455). The TiO₂ crystals with size of 30–85 nm only exposed 18% {001} facets, but they showed an increase in the specific surface area as 21 m² g⁻¹, and exhibited 5.6 times stronger photoactivity than microcrystals with 72% {001} facets (*Chem. Commun.* 2010, **46**, 755).

Thus there should be a balance between the particle size and percentage of exposed {001} facets. The study on photocatalytic CO₂ conversion indicated that the TiO₂ crystal with 60% {001} facets (60 nm) showed 15% and 90% higher CO production than that with 92% {001} facets (150 nm) and 95% {101} facets (20 nm), respectively (*ACS Catal.* 2016, **6**, 1097). Yu et al. systematically studied the influence of factors on the photocatalytic CO₂ conversion including percentage of {001} facets, size, and surface area (*J. Am. Chem. Soc.* 2014, **136**, 8839). The table below shows that the highest CH₄ production was achieved by the TiO₂ crystals with 58% {001} facets, size of 60 nm, and surface area of 45 m² g⁻¹ (**Table R1**).

Table R1. Comparison of the percentage of {001} facets, size, specific surface area, and photocatalytic CO₂ conversion of the samples (*J. Am. Chem. Soc.* 2014, **136**, 8839).

sample	{001} facets %	size (nm)	surface area (m ² g ⁻¹)	CH ₄ production (μmol g ⁻¹ h ⁻¹)
HF0	11	13	107	0.15
HF3	49	-	-	0.75
HF4.5	58	80	45	1.35
HF6	72	-	-	0.82
HF9	83	100	30	0.55
P25	-	-	-	0.38

4. In thermogravimetric analysis (TGA, Figure S9), the authors have stated that the HCP-TiO₂-FG composite structure exhibits excellent thermal stability compared to TiO₂-G with resistance of degradation up to 400 °C. However, according to Figure S9, the thermal stability of TiO₂-G is even better than HCP-TiO₂-FG with resistance of degradation up to 400 °C.

Response: Sorry for this mistake. We have revised the sentence on page 6 as follows, “*the HCP-TiO₂-FG composite structure exhibited the excellent thermal stability comparable to TiO₂-G with resistance to degradation up to 400 °C*”.

5. For the photocatalytic test, the authors should provide the details for the optical density of light source and the illuminated area.

Response: The details for the photocatalytic test are as follows: Under visible-light ($\lambda \geq 420$ nm) irradiation, the optical density of 300 W Xe lamp was measured to be 433 mW cm⁻² and the illuminated area of photocatalyst is about 3.14 cm².

In the revised manuscript, the following information is added,

The optical density was measured to be 433 mW cm⁻² and the illuminated area of photocatalyst was about 3.14 cm².”

In addition, to verify the origins of produced CO and CH₄, the isotopic ¹³CO₂ must be used as a reactant to trace the carbon sources in the photocatalytic reaction.

Response: As suggested, to verify the origin of the products, we have conducted the isotopic labeling experiment using ¹³CO₂ as a reactant to trace the carbon sources in the

photocatalytic reaction. The results are shown in **Supplementary Fig. S15**.

We have also included corresponding description on page 9 in the manuscript as follows:

“To verify the evolution of CO and CH₄ from CO₂ conversion over HCP-TiO₂-FG photocatalyst, we conducted three controlled experiments: (1) irradiation of catalyst under inert N₂ condition; (2) the use of isotopic labeled ¹³CO₂ and H₂¹⁸O as the reactants; (3) irradiation of catalyst in the presence of CO₂ gas without H₂O vapors.”

“In an isotopically labeled experiment, the ¹³CH₄ and ¹³CO signals at m/z = 17 and m/z = 29 appeared after the photocatalytic reaction. The results confirmed that the CO and CH₄ products are indeed originating from the photocatalytic reduction of CO₂ gas (Supplementary Fig. S15).”

Figure S15. GC-MS spectra of ¹³CH₄ (m/z=17) and ¹³CO (m/z=29) after the photocatalytic reaction over HCP-TiO₂-FG under visible-light irradiation ($\lambda \geq 420$ nm). The isotopic labeled ¹³CO₂ was used as the substrate in normal experiment condition.

6. According to Figure 4a and b, the HCP-FG material also exhibits a photocatalytic activity for CO and CH₄ production. The authors have stated that the catalytic sites are located on TiO₂. In this case, it remains elusive whether the catalytic sites are located on TiO₂ or HCP-FG. The authors should clarify this point.

Response: Yes, the HCP-FG material also exhibits a photocatalytic activity for CO and CH₄ production. The porous property and photocatalytic performance of samples are shown in **Fig. 3d-f**, **Fig. 4a-c**, **Fig. S10**, **Fig. S12**, **Table 1**, and **Table S2**. To clarify the location of catalytic sites, the porous property and photocatalytic performance of HCP-FG, HCP-TiO₂-FG, and TiO₂/HCP-FG samples are selected and shown in the **Fig. R1** and **Table R2** below.

On page 8-9, we have added the corresponding discussions as follows:

“The adsorptive and catalytic sites can be clarified through the comparison in porous property and photocatalytic performance. Obviously, the introduction of porous HCPs layers enriched the adsorptive sites to achieve the high CO₂ uptake and improved the

visible light absorption. Thus the formation of well-defined HCP-TiO₂-FG sandwich structure resulted in much higher photocatalytic CO₂ reduction rate.”

“Although TiO₂ deposition blocked most of the adsorptive sites of HCP-FG and resulted in a dramatic decrease to less than one-third of CO₂ uptake, while the CH₄ production over TiO₂/HCP-FG is 7.4 times more than that over pristine HCP-FG. The comparison between HCP-FG and HCP-TiO₂-FG shows that the catalytic sites on TiO₂ were much more active for CO₂ reduction than those on HCP-FG.”

Figure R1. (a) Nitrogen adsorption and desorption isotherms at 77.3 K of samples. (b) Volumetric CO₂ adsorption isotherms and desorption isotherms up to 1.00 bar at 273.15 K of samples. (c) Pore size distribution that calculated using DFT methods (slit pore models, differential pore volumes). Time-dependent production of CH₄ (d) and CO (e) in photocatalytic CO₂ reduction with different catalysts under visible-light ($\lambda \geq 420$ nm). (f) Average efficiency of photocatalytic CO₂ conversion with different catalysts during 5 h of visible-light ($\lambda \geq 420$ nm) irradiation.

Table R2. Comparison of the selected samples in porous property and photocatalytic performance.

Sample	S _{BET} ^a (m ² g ⁻¹)	CO ₂ uptake ^b (wt %)	PV ^c (cm ³ g ⁻¹)	Visible-light irradiation (μmol g ⁻¹ h ⁻¹)		
				r(CH ₄) ^d	r(CO) ^e	R _e ^f
HCP-FG	593	10.90	0.380	0.90	7.62	22
TiO ₂ /HCP-FG	178	3.31	0.164	6.71	16.17	86
HCP-TiO ₂ -FG	988	12.87	0.693	27.62	61.63	264

^a Surface area calculated from nitrogen adsorption isotherms at 77.3 K using BET equation. ^b CO₂ uptake determined volumetrically using a Micromeritics ASAP 2020 M analyzer at 1.00 bar and

273.15 K. ^c Pore volume calculated from nitrogen isotherm at P/P₀=0.995, 77.3 K. ^{d,e} average of gas evolution rate (r) during 5 h of photocatalytic CO₂ reduction. ^f R_e is the rate of total consumed electron number for the reduced product; R_e = 8r(CH₄)+2r(CO).

7. In Figure 4f, the authors have used the band gap energy, HOMO and LUMO levels of HCP-FG to discussing the charge separation and transfer within the HCP-TiO₂-FG sandwich photocatalyst. This analysis is misleading. According to the sandwiched structure of HCP-TiO₂-FG, the photogenerated electrons migrate from HCP to TiO₂ via their interfacial interaction. As a result, the authors should use the band gap energy, HOMO and LUMO levels of HCP rather than HCP-FG in the analysis.

Response: We appreciate the reviewer's kind suggestion. We would like to make an explanation on the structure of HCP-TiO₂-FG here. The HCPs and graphene are not free-standing in the sandwiched structure. Benefiting from the *in-situ* knitting strategy, ultrathin HCPs layers are integrated with the functionalized graphene nanosheets through the methylene linkers, and the structures have been verified by XPS, FT-IR, and CP/MAS NMR characterizations (**Fig. 3b-c** and **Supplementary Fig. S6-8**). Thus, the electrons can be moved throughout the whole HCP-FG structure owing to their covalent linking. Under visible-light irradiation, the electrons were generated in the integrated HCP-FG structure and then migrated to TiO₂ *via* their interfacial interaction.

8. According to the time-dependent production of CH₄ and CO as shown in **Figure S13**, it remains unclear why the production of CH₄ increases linearly while the production of CO has a stagnation effect.

Response: Yes, the CH₄ production increased linearly with irradiation time, whereas the CO production was fast at initial irradiation time and then showed a sluggish increase during the photocatalytic reaction (**Fig. 4a-b** and **Fig. S19**). In fact, the stagnation effect in CO production has also observed in photocatalytic CO₂ reduction by many researchers (*J. Am. Chem. Soc.* 2018, **140**, 38; *J. Am. Chem. Soc.* 2017, **139**, 7217; *J. Am. Chem. Soc.* 2017, **139**, 6538; *Adv. Mater.* 2016, **28**, 6485; *ACS Nano*, 2015, **9**, 2111). For example, Luo's group studied the mechanism of photoreduction of CO₂ to CH₄ on TiO₂ surface by theoretical calculations. They proposed that CO was the initial product of CO₂ photoreduction that could be further photo-reduced to CH₃OH or CH₄ (*ACS Catal.* 2016, **6**, 2018). That is possibly why the CO production rate is fast at initial irradiation time and then shows a sluggish increase during the photocatalytic

reaction. In contrast, the CH₄ as the final product showed a steady increase.

9. *In the electrochemical impedance spectra (EIS, Figure S15), HCP-FG has a much larger semi-circle diameter than HCP-TiO₂-FG. Why the addition of TiO₂ can dramatically reduce the electron-transfer resistance?*

Response: In the EIS analysis (**Supplementary Fig. S23**), the smaller arc in HCP-TiO₂-FG sandwich structure than HCP-FG suggests that the formation of sandwich structure improves the electronic conductivity. According to the gas adsorption-desorption analysis, it was found that the surface area and CO₂ uptake of HCP-TiO₂-FG were higher than those of HCP-FG (**Supplementary Tab. S2**). Thus we can infer that the TiO₂ intercalation somewhat restricted the aggregation of HCPs layers on graphene nanosheets resulting in the reduced electron-transfer resistance.

10. *In abstract, "...and highlights the importance of MOPs in combination with photocatalysts for solar energy conversion.", there is no definition for the "MOPs" in the manuscript.*

Response: We are sorry for missing the definition of the "MOPs". In the revised manuscript, we have replaced it by "microporous organic polymers".

Reviewer 3:

The authors report on a novel catalyst for photocatalytic CO₂ activation using visible light enabling unprecedented activity and CH₄ selectivity. This high performance is achieved by tailoring the CO₂ adsorption of the material together with short diffusion length to the active sites.

To achieve these features, the authors prepare a composite material composed of TiO₂ anatase crystals with exposed {001} facets as active sites grown on graphene oxide. The latter was functionalized by PhPh₃ serving as anchoring groups enabling coating of a thin layer of hypercrosslinked polymer (HCP).

The material is comprehensively characterized. Its catalytic activity is reported compared to pure titania as well as further references such as the composite without titania, titania supported on graphene oxide-HCP composite etc. The results are presented in a systematic and sound manner.

Concerning the impact, the contribution is clearly important but justification for a publication in Nature Communications is not sufficiently provided. In addition, the hypothesis of the contribution is based on CO₂ adsorption and diffusion length as

crucial parameters to enhance catalytic activity.

Response: First, we would like to thank the reviewer to realize and appreciate the importance of work reported in this manuscript. We also thank the reviewer's overall comments and kind suggestions that helped us to improve the manuscript significantly. We have now addressed all the reviewer's query point-wise in our response below.

Following this line of argument, the major aspects to critically consider are:

- CO₂ adsorption on or close to the active site is important, as surface coverage presents the concentration subsequently determining the reaction rate to be achieved ($r=k*\text{surface coverage of reagents}$, considering a Langmuir Hinshelwood type of activation). In line, a higher surface coverage of the substrates enables reaching a higher reaction rate, while the intrinsic catalytic activity of the active site remains unaltered. Following this argument, the authors have to carefully proof their hypothesis that CO₂ adsorption within the material indeed causes enhanced coverage on or close to the active sites, increasing reaction rate. Kinetic experiments varying partial pressure, temperature etc. and providing assessment of the influence on the reaction rate are indispensable to justify the hypothesis. In addition, a careful representation of the literature state of the art including rates achieved in previous contributions normalized to the active sites content of the different catalysts are essential.

Response: This suggestion inspired us to investigate the kinetic characteristics of the photocatalytic CO₂ conversion which has never been discussed in the related literatures to the best of our knowledge. Generally, the reaction kinetics in a gas-solid system is studied by varying the partial pressure and temperature.

The photocatalytic reactions were carried out in a batch system under standard atmospheric pressure. Since the pressure in the photocatalytic reactor was settled as standard atmospheric pressure, we conducted the kinetic experiments by varying the partial pressure of CO₂. The HCP-TiO₂-FG photocatalyst exhibits a high CO₂/N₂ selectivity ratio of 25.8 calculated by the initial slopes of adsorption isotherms shown in **Supplementary Fig. S20** (*Adv. Mater.* 2012, **24**, 5703). The partial pressure of CO₂ can be adjusted from 2.5% to 100% by varying the volume ratio of CO₂ to N₂. Since the kinetic model and reaction mechanism of photocatalytic CO₂ conversion are ambiguous so far, the quantitative relationship between CO₂ coverage and CH₄

evolution rate is still unclear. Interestingly, it is observed that they show a similar trend of increase with CO₂ proportion, e.g. both showed a dramatic increase at lower partial pressure and then displayed a sluggish increase at higher CO₂ concentration (**Supplementary Fig. S21**).

As far as temperature-dependent kinetics is concerned, the results and related discussion given below in our response would be helpful to address this comment.

In addition, the literature state of the art including rates normalized to the active sites content of the different catalysts are reviewed in the revised manuscript. Based on the discussions, we can deduce that the more efficient CH₄ production over HCP-TiO₂-FG should not result from the difference in the number of catalytic sites but mostly come from the higher surface coverage of CO₂ on the active sites. Thus we can furthermore demonstrate the superiority of such porous sandwich structure towards the visible-light-driven photocatalytic CO₂ conversion.

On page 10, we have added the following content:

*“The kinetics experiments were established to explore the relationship between the CO₂ adsorption and CH₄ production. The surface coverage of CO₂ on the active sites can be varied by adjusting the partial pressure of CO₂ in CO₂/N₂ mixed gas because of a high CO₂/N₂ selectivity ratio of 25.8 over the HCP-TiO₂-FG photocatalyst (**Supplementary Fig. S20**). Since the kinetic model and reaction mechanism of photocatalytic CO₂ conversion are ambiguous so far, the quantitative relationship between CO₂ coverage and CH₄ evolution rate is still unclear. Interestingly, it is observed that they show a similar trend of increase with CO₂ proportion, e.g. both of them dramatically increased at lower partial pressure and then displayed a sluggish increase at higher CO₂ concentration (**Supplementary Fig. S21**). Generally, the reaction rates that normalized to the active sites allow the direct comparison of intrinsic reactivity on different catalysts⁴⁷⁻⁴⁹. For the catalytic system employing same catalyst, the reaction rate appears to be independent of the loading amount of catalyst after normalization to the same amount^{50, 51}. In this regard, the porous HCP-TiO₂-FG photocatalyst possesses equivalent catalytic active sites to TiO₂/HCP-FG hybrid due to the same content of TiO₂ photocatalyst. That is, the more efficient CH₄ production over*

HCP-TiO₂-FG should not result from the difference in the number of catalytic sites but mostly come from the higher surface coverage of CO₂ on the active sites.”

Figure S20. *CO₂ and N₂ adsorption isotherms of porous HCP-TiO₂-FG sandwich structure at 273 K. The HCP-TiO₂-FG photocatalyst exhibits a high CO₂/N₂ selectivity ratio of 25.8 calculated by the initial slopes of adsorption isotherms³⁷.*

Figure S21. *Influence of the partial pressure of CO₂ on the CO₂ uptake (a) and CH₄ production rate (b). The photocatalytic reactions were carried out in a batch system under standard atmospheric pressure. The partial pressure of CO₂ can be adjusted from 2.5% to 100% by varying the volume ratio of CO₂ to N₂. q_e is the equilibrium adsorption capacity at pure CO₂ atmosphere with 1 bar. q/q_e represents fractional uptake at different partial pressure of CO₂.*

The second argument relates to diffusion length as important element of catalyst design. The provided data do not allow any conclusion of the role of surface diffusion within

the overall system. Following the argument of a diffusion governed process, the rate of surface diffusion should be somewhat rate determining. Consequently, kinetic analysis following e.g. the temperature dependence are needed to support such a hypothesis. Diffusion limitation should cause a limited apparent activation energy according to the reduced temperature dependence of various types of diffusion compared to chemical reactions.

Response: As suggested, we have conducted the kinetic experiments by varying the temperature on the photocatalyst surface. The temperature has a complicated influence on the rate of photocatalytic conversion from the aspects of adsorption, diffusion and photocatalytic processes.

The results are shown in **Fig. 3e-f** and **Fig. R2-3**. By elevating the temperature, the surface coverage of CO₂ molecule on the catalyst surface was decreased due to the exothermic effect of adsorption process (**Fig. 3e-f**), while the diffusion rate was increased as a result of the increased thermal motion of CO₂ molecules (**Fig. R2**). For the photocatalytic process, it is well-known that the Gibbs free energy increases in the photocatalytic conversion of CO₂ to CO and CH₄. The increase of chemical potential originates from the energy of photons rather than heat, so the change in temperature by dozens of degrees would not cause a measurable difference in the photocatalytic reaction. That is why the activation energy is seldom discussed in the photocatalytic reaction in literatures. From our experience, the photocatalytic CO₂ conversion efficiency over pure TiO₂ photocatalyst under UV-light is indeed independent of temperature from 291 K to 353 K. Interestingly, we observed an obvious increase in CH₄ evolution rate over porous HCP-TiO₂-FG photocatalyst with temperature (**Fig. R3**).

Now we can conclude as follows: by increasing the temperature, the surface coverage of CO₂ molecule declined, the diffusion rate speeded up, the photocatalytic reaction rate kept constant, and the overall reaction rate for CO₂ conversion was increased. Based on Arrhenius plot, the adsorption activation energy for CO₂ adsorption is calculated to be 5.20 kJ mol⁻¹ (**Fig. R3a**) using a microporous diffusion model (*Ind. Eng. Chem. Res.* 2009, **48**, 10015-10020; *J. Am. Chem. Soc.* 2004, **126**, 1356-1357). Since the CH₄ production increases linearly and possesses dominant electron consumption selectivity as 83.7%, we can adopt the pseudo-zero order model to estimate the rate constant for the overall reaction, obtaining apparent activation energy

as 9.34 kJ mol^{-1} (**Fig. R3b**). By noting the temperature-dependent characteristics with activation energy values, we can conclude that the rate of surface diffusion is somewhat rate determining to the photocatalytic CO_2 conversion.

In the present study, we aim to provide new insights into the design and synthesis of well-defined porous photocatalysts for CO_2 uptake and conversion, and present an important first example towards solar-to-carbonaceous fuels conversion employing microporous organic polymers in combination with photocatalytic materials. The kinetic analysis is indeed very important to the study and applications of photocatalytic CO_2 conversion but still there are many challenges that need to be addressed to unveiling the fundamental understanding of each reaction step. Therefore, we now intend to adopt the simple photocatalytic system such as pure TiO_2 photocatalyst to probe the kinetics model and reaction mechanism of CO_2 conversion, and then extend to complicated models involving adsorption, diffusion and photocatalytic processes.

Figure R2. (a, b, c, d) Adsorption kinetic curves of HCP-TiO₂-FG at different temperature. (e) Fractional adsorption uptake (q/q_e) at different temperature. (f) Plots of the fractional adsorption uptake (q/q_e) against the square root of adsorption time at different temperature. The diffusion coefficient D_M is calculated using a microporous diffusion model: $\frac{q_t}{q_e} \cong \frac{6}{r_c} \sqrt{\frac{D_M}{\pi}} \sqrt{t}$. r_c is the average thickness of the HCP layer. (*Ind. Eng. Chem. Res.*, 2009, **48**, 10015-10020)

Figure R3. Arrhenius plot of CO₂ diffusivity (a) and CH₄ production rate (b) over HCP-TiO₂-FG sandwich structure.

- As a minor point, the experimental data are not provided with sufficient information to understand the used experimental setup, e.g. in the related figure captions. Where the reactions carried out in a flow reactor or a batch system? What was the temperature? Partial pressure of CO₂ and H₂O were constant (was there water in the system)?

Response: The photocatalytic reactions were carried out in a batch system under standard atmospheric pressure. The partial pressure of CO₂ and H₂O were constant with the existence of water below the scaffold loading photocatalyst. Under visible-light irradiation, the temperature of the water was measured to be about 50 °C. The detailed information has been added in the caption of Fig. 4.

- Concerning CO₂ adsorption experiments, data in presence of water appear important and water vapor sorption experiments would be complementary.

Response: Following the reviewers' suggestions, we measured the water vapors sorption capacity and the CO₂ uptake in the presence of water vapors respectively. The results are shown in Supplementary Fig. S17-18.

On page 9, we have added the following content:

“Although the porous HCP-TiO₂-FG also exhibits a high adsorption capacity towards water vapors, about 30 wt% at 90% humidity (Supplementary Fig. S17), the existence of water vapors brings a slight increase in CO₂ uptake (Supplementary Fig. S18), presumably due to their affinity with the water molecules.”

Figure S17. The water adsorption of HCP-TiO₂-FG at different humidity.

Figure S18. CO₂ adsorption experiments, data in presence of water. q_e is the equilibrium adsorption capacity at pure CO₂ atmosphere with 1 bar. q/q_e represents fractional uptake at pure CO₂ atmosphere and mix atmosphere of 99% CO₂+1% water vapors with 1 bar.

Reviewers' comments:

Reviewer #1 (Remarks to the Author):

The authors reasonably answered to my comments and properly revised the paper. If the other reviewers also agree to accept, the revised version is publishable in Nature Communications.

Reviewer #2 (Remarks to the Author):

Comments:

I am pleased to see that the authors have made a big effort to improve the quality of this manuscript by performing new experiments, adding new discussion and completing a number of changes. Indeed, some comments have been partially addressed. However, the evidences are still insufficient to support their key conclusions. Although it is a topic of general interest, the current revised manuscript still cannot meet the level for publication in Nature Commun.

Specific comments are listed below:

1. In the isotopically labeled experiment, why the $^{13}\text{CH}_4$ and ^{13}CO signals are only at $m/z = 17$ and $m/z = 29$, respectively? Why there are no fragment signals at $m/z = 13$ and $m/z = 16$ for $^{13}\text{CH}_4$ and ^{13}CO ?
2. As the authors stated that the catalytic sites are located on TiO_2 , the functions of the formed HCP- TiO_2 -FG sandwich structure remain not clear. In the sandwich structure, what are the TiO_2 crystals directly contacting with (the porous hypercrosslinked polymer (HCP) or graphene)? If the TiO_2 crystals are directly contacting with the HCP, the photogenerated electrons will migrate from HCP to TiO_2 via their interfacial interaction. In this case, how can graphene improve the charge separation efficiency? If the TiO_2 crystals are directly contacting with the graphene, the function of porous HCPs layers in enriching the adsorptive sites to achieve the high CO_2 uptake may not work.
3. In the CV measurement for calculating the HOMO and LUMO energy levels of HCP- TiO_2 -FG catalyst in CH_2Cl_2 , the reference electrode Ag/AgCl is not correct. The reference electrode Ag/AgCl is commonly suitable in aqueous solution. In organic solvent, the reference electrode should be Ag/Ag^+ using the Fc/Fc^+ couple as an internal standard.
4. The authors demonstrated that ultrathin HCPs layers were integrated with the functionalized graphene nanosheets through the methylene linkers. However, there is no direct evidence to conclude that HCPs layers were integrated with the graphene nanosheets through the methylene linkers. The current data can only support that there exists the methylene.

Reviewer #3 (Remarks to the Author):

The authors were asked to provide further evidence for the governing nature of their two main Arguments, namely the superior CO_2 adsorption as pre-requisite of high substrate concentration close to or on the active site and their hypothesis on the shortend diffusion lengths.

Indeed, a suitable kinetic analysis has been carried out illustrating that the investigated photocatalytic

CO₂ reduction is not under intrinsic kinetic control of the catalyst but the observed rates of CO and CH₄ formation are rather determined by some transport effects.

The observed limited dependence of rate on temperature may hint towards diffusion control. I fully agree that kinetic analyses is a yet under-represented aspect in photo-catalysis. Though, the little rate dependence on temperature appears to point towards film diffusion effects. Has the stirring speed or the main particle size been varied?

It remains unclear for me why the authors conclude on a reduced Diffusion length for the Optimum System, although they do not know which Diffusion is rate limiting: CO₂ from bulk to the film, through the film, in/on the porous material or Charge carrier diffusion?

The authors made an effort to identify the corresponding oxidation Reaction which has to balance the observed reduction reactions. Oxygen has been proven but quantitative Analysis is not yet available. What about oxidation of the formed products, e.g. CO and CH₄ to CO₂ as reverse reactions of the attempted CO₂ reduction. I suggest reference Experiments with these Substrates.

Overall, the contribution may become suitable for Nature Communications after providing further evidence and comments to the points made before.

Response to Reviewers' Comments

Many thanks for forwarding us the reviewers' comments that have helped us a lot to significantly improve the manuscript. Below are our responses (in BLUE colour) to the editor's queries.

Reviewer 2:

1. In the isotopically labeled experiment, why the $^{13}\text{CH}_4$ and ^{13}CO signals are only at $m/z = 17$ and $m/z = 29$, respectively? Why there are no fragment signals at $m/z = 13$ and $m/z = 16$ for $^{13}\text{CH}_4$ and ^{13}CO ?

Response: The fragment signals at $m/z = 13$ and $m/z = 16$ were not observed in GC-MS spectra due to the concentrated region of horizontal axis in **Supplementary Fig. S15** in the previous version. The region of horizontal axis is now extended to display the ^{13}C and ^{16}O fragments of $^{13}\text{CH}_4$ and ^{13}CO in the updated **Supplementary Fig. S15**.

Figure S15. GC-MS spectra of gas products after the photocatalytic reaction over HCP-TiO₂-FG under visible-light irradiation ($\lambda \geq 420$ nm). The isotopically labeled $^{13}\text{CO}_2$ was used as a substrate under normal experimental conditions.

2. As the authors stated that the catalytic sites are located on TiO₂, the functions of the formed HCP-TiO₂-FG sandwich structure remain not clear. In the sandwich structure, what are the TiO₂ crystals directly contacting with (the porous hypercrosslinked polymer (HCP) or graphene)? If the TiO₂ crystals are directly

contacting with the HCP, the photogenerated electrons will migrate from HCP to TiO_2 via their interfacial interaction. In this case, how can graphene improve the charge separation efficiency? If the TiO_2 crystals are directly contacting with the graphene, the function of porous HCPs layers in enriching the adsorptive sites to achieve the high CO_2 uptake may not work.

Response: We would like to point out that the HCPs and graphene existed as an integrated structure instead of freestanding parts in the composite. Benefiting from the *in-situ* knitting strategy, ultrathin HCPs layers were integrated with the functionalized graphene nanosheets through the methylene linkers. The specific structure of HCP- TiO_2 -FG has been clearly illustrated at Page 3 in the manuscript and then verified by XPS, FT-IR, and CP/MAS NMR characterizations (**Fig. 3b-c** and **Supplementary Fig. S6-S8**). The covalent linking with graphene effectively improves the electronic conductivity of the HCPs and thus facilitates the electron transfer in the composite (**Supplementary Fig. S21**). Thus the electrons can be moved throughout the whole HCP-FG structure owing to their covalent linking. Under visible-light irradiation, the electrons were generated in the integrated HCP-FG structure and then migrated to TiO_2 *via* their interfacial interaction. Based on the results obtained, we have presented better understanding of such superior HCP- TiO_2 -FG structure from the aspects of improving CO_2 adsorption and diffusion, visible-light absorption, and photogenerated charge separation efficiency.

In the revised manuscript, more explanations on the HCP- TiO_2 -FG structure are presented in the response to Comment 4#. The diagram in Figure 1 has been updated to show the connections between HCPs and functionalized graphene. We are hoping that the structure is now much clear and clearly understandable.

Figure 1. Construction of a well-defined porous HCP- TiO_2 -FG sandwich structure. (I) The functionalization of TiO_2 -G by diazonium salt formation. (II) The knitting of TiO_2 -FG with

syn-PhPh₃ by solvent knitting method.

3. In the CV measurement for calculating the HOMO and LUMO energy levels of HCP-TiO₂-FG catalyst in CH₂Cl₂, the reference electrode Ag/AgCl is not correct. The reference electrode Ag/AgCl is commonly suitable in aqueous solution. In organic solvent, the reference electrode should be Ag/Ag⁺ using the Fc/Fc⁺ couple as an internal standard.

Response: For the CV measurement in organic solvent, Ag/AgCl is also the widely used reference electrode besides Hg/Hg₂Cl₂ and Ag/Ag⁺ (*Nat. Mater.* **2017**, 16, 220; *Adv. Mater.* **2017**, 29, 1606054; *Angew. Chem. Int. Ed.* **2017**, 56, 13503; *Adv. Mater.* **2018**, 30, 1706124; *Adv. Energy Mater.* **2018**, 8, 1702251). The energy level of the Ag/AgCl reference electrode can be calibrated with ferrocene/ferrocenyl couple (Fc/Fc⁺).

According to the suggestion, the CV measurement using Ag/Ag⁺ as reference electrode was also performed to ensure the identical electrolyte in the system. The CV curves were updated in Figure S23 in the revised supplementary information. The calculated HOMO and LUMO levels vs. vacuum level are the same as the previous values.

Figure S23. Cyclic voltammetry (CV) measurements of ferrocene (FC) reference and HCP-TiO₂-FG catalyst on ITO glass electrode as the working electrode. The CV curves were recorded in N₂-saturated anhydrous acetonitrile containing 0.1 M tetrabutyl-ammonium hexafluorophosphate (Bu₄NPF₆). The ITO glass electrode with no catalyst coating was labeled as blank for comparison. Reference electrode: Ag/Ag⁺ (0.01 M of AgNO₃ in acetonitrile). Counter electrode: Pt wire. Scan rate: 100 mV/s.

4. The authors demonstrated that ultrathin HCPs layers were integrated with the functionalized graphene nanosheets through the methylene linkers. However, there is no direct evidence to conclude that HCPs layers were integrated with the graphene

nanosheets through the methylene linkers. The current data can only support that there exists the methylene.

Response: Many graphene-polymer composite materials are usually based on functionalized graphene which act as 2D template, then the reactive monomers are added and the polymer structures constructed on the graphene surface. (*Angew. Chem. Int. Ed.* **2013**, 52, 9668; *Adv. Mater.* **2014**, 26, 3081; *Angew. Chem. Int. Ed.* **2015**, 54, 1812; *Angew. Chem. Int. Ed.* **2018**, 57, 1034.) The chemical structures of composite materials are generally undefined and these papers usually show the approximate chemical structure. It could be confirmed that the polymers are attached to graphene surface by characterizations of SEM, HR-TEM and AMF. In ours case, the growth of HCP layer on TiO₂-FG can also be determined by SEM, HR-TEM and AFM. We have tried to explain the relation between the HCP structure and TiO₂-FG by a model experiment.

Model experiment process: the solvent knitting process was performed as follows: Under N₂ atmosphere, the catalyst (AlCl₃, 104 mg) was added to TiO₂-FG (40 mg) in 8 mL dichloromethane (without monomer PhPh₃), and then the system connected to nitrogen-line to form relatively less air-controlled environment. The reaction system was then stirred at 0 °C for 4 h, 30 °C for 8 h, 40 °C for 12 h, 60 °C for 12 h and 80 °C for 24 h to obtain the resulting material. The resulting precipitate was quenched using HCl-H₂O (v/v = 2:1), and washed thrice with water and twice with ethanol, extracted with ethanol for 48 h, and finally dried in a vacuum oven at 65 °C for 24 h.

Figure R1 ¹³C CP/MAS NMR spectra of TiO₂-FG (a) and HO-CH₂-TiO₂-FG (b).

Fig. R1 shows the ^{13}C CP/MAS NMR curves of $\text{TiO}_2\text{-FG}$ and $\text{HO-CH}_2\text{-TiO}_2\text{-FG}$ obtained from the model experiment procedure. Compared with **Fig. R1a** and **R1b**, the peak of C at 137 ppm is obviously enhanced, which indicates that the benzene ring in $\text{TiO}_2\text{-FG}$ has electrophilic substitution reaction with dichloromethane (DCM). It is noteworthy that the peaks belong to methylene at 37-39 ppm, which were commonly seen in HCPs materials and disappeared in **Fig. R1a** (*Macromolecules* **2011**, 44, 2410; *Sci. Adv.* **2017**, 3, e1602610). Instead, a new peak at 62.5 ppm was shown in **Fig. R1b**, which was corresponding to the methylene C peak of benzyl alcohol group ($-\text{CH}_2\text{-OH}$). This indicates that the phenyl groups of $\text{TiO}_2\text{-FG}$ change into benzyl chloride group ($-\text{C}_6\text{H}_4\text{-CH}_2\text{-Cl}$) in the reaction process firstly (**Fig. R2a**). However, due to the barrier effect of TiO_2 nanoparticles on graphene sheets, benzyl chloride groups can only exist in the form of original states instead of crosslinking with other benzene rings. With the quenching of dilute hydrochloric acid, the active benzyl chloride groups react with water and formed the benzyl alcohol groups ($-\text{C}_6\text{H}_4\text{-CH}_2\text{-OH}$) (**Fig. R2b**). The model experiment indicates that the phenyl groups on $\text{TiO}_2\text{-FG}$ can react electrophilically with DCM to form benzyl chloride groups ($-\text{C}_6\text{H}_4\text{-CH}_2\text{-Cl}$), which cannot undergo the further crosslinking reaction due to the blocking effect of TiO_2 nanoparticles. Meanwhile, the specific surface area of $\text{HO-CH}_2\text{-TiO}_2\text{-FG}$ ($128\text{ m}^2\text{ g}^{-1}$) obtained from the model experiment did not change compared with $\text{TiO}_2\text{-FG}$, which also prove the above-mentioned conclusion.

Figure R2 Schematic diagram of reaction process in model experiment.

As co-monomer of synthesis for $\text{HCP-TiO}_2\text{-FG}$, *syn*- PhPh_3 can be self-crosslinked to obtain SHCP-3a with a high specific surface area of $2525\text{ m}^2\text{ g}^{-1}$ by solvent knitting method (*Sci. Adv.* **2017**, 3, e1602610). If $\text{TiO}_2\text{-FG}$ and SCHP-3a were mixed simply according to same mass ratio of co-monomers, the specific surface area and CO_2 uptake amount of the mixtures were lower than these of $\text{HCP-TiO}_2\text{-FG}$,

which suggest two monomers form a homogenous "sandwich" structure (**Tab. R1**). Meanwhile, no freestanding SHCP-3a blocks were observed in SEM, TEM and scanning transmission electron microscopy (STEM) images (**Fig. 2c-f and Supplementary Fig. S1d**). This is because benzyl chloride group (-C₆H₄-CH₂-Cl) formed has high reactivity for TiO₂-FG and was linked to co-monomer *syn*-PhPh₃, and the "sandwich" structure was formed in which the HCPs porous organic layers and TiO₂-FG were linked by methylene.

Table R1 The relationship of the surface area and CO₂ uptake of the resulting materials and the amount of *syn*-PhPh₃.

No.	Mass ratio ^a	S _{BET} ^b (m ² g ⁻¹)	S _{BET} ^c (m ² g ⁻¹)	CO ₂ uptake ^d (wt %)	CO ₂ uptake ^e (wt %)
TiO ₂ -FG	0	131	131	3.11	3.11
HCPs-TiO ₂ -FG-1	0.2	530	600	6.1	7.71
HCPs-TiO ₂ -FG	0.5	929	988	9.0	12.87
HCPs-TiO ₂ -FG-3	1	1328	1206	12	13.71
SHCP-3a	+∞	2525	2525	20.9	20.9

^a Mass ratio of co-monomers for *syn*-PhPh₃ and TiO₂-FG. ^b the sum of surface area for SHCP-3a and HCPs-TiO₂-FG-X with mass ratio of co-monomers. ^c Surface area calculated from nitrogen adsorption at 77.3 K using Langmuir equation. ^d the sum of CO₂ uptake for SHCP-3a and HCPs-TiO₂-FG-X with mass ratio of monomers at 1.00 bar and 273.15 K. ^e CO₂ uptake determined volumetrically using a Micromeritics ASAP 2020 M analyzer at 1.00 bar and 273.15 K.

Reviewer 3:

The authors were asked to provide further evidence for the governing nature of their two main Arguments, namely the superior CO₂ adsorption as pre-requisite of high substrate concentration close to or on the active site and their hypothesis on the shortend diffusion lengths. Indeed, a suitable kinetic analysis has been carried out illustrating that the investigated photocatalytic CO₂ reduction is not under intrinsic kinetic control of the catalyst but the observed rates of CO and CH₄ formation are rather determined by some transport effects. The observed limited dependence of rate

on temperature may hind towards diffusion control. I fully agree that kinetic analyses is a yet under-represented aspect in photo-catalysis. Though, the little rate dependence on temperature appears to point towards film diffusion effects. Hast the stirring speed or the main particle size been varied? It remains unclear for me why the authors conclude on a reduced Diffusion length for the Optimum System, although they do not know which Diffusion is rate limiting: CO₂ from bulk to the film, through the fild, in/on the porous material or Charge charrier diffusion?

Response: Many thanks for appreciating our efforts on the kinetic analysis presented in the previous revision. The photocatalytic reactions were carried out in a gas-solid batch system without stirring the gas mixture. According to the suggestion, we studied the diffusion process by varying the stirring speed from zero to maximum. As shown in **Supplementary Fig. S33**, the increase in stirring speed greatly facilitates the photocatalytic conversion of CO₂ to CH₄ product, indicating that the diffusion plays a significant role in such a gas-solid reaction system. The related results and discussions have been added as a separate section “kinetic analysis” in the revised manuscript.

The particle size is another factor affecting the diffusion process, especially for internal diffusion. We would like to mention that the HCP-TiO₂-FG composite produced through this strategy possesses the specific particle size. In the experience of optimizing synthesis conditions, TiO₂ crystals with larger or smaller size are inclined to form agglomerates, which cannot uniformly decorate on graphene or be fully wrapped by ultrathin HCPs layers. In the current system, we cannot achieve the varied particle size while keeping the designed structure with similar surface area, CO₂ uptake capacity and the number of active sites.

In order to reduce the diffusion length, we tried to construct the HCP-TiO₂-FG sandwich structure with ultrathin HCP layers. The reduced diffusion length should be beneficial for the diffusion of CO₂ and photo-generated charges to the catalytically active sites, both of which may contribute to the improved CO₂ conversion efficiency. Based on the gas diffusivity measurement, the diffusion coefficient of CO₂ in the HCP-TiO₂-FG sandwich structure is calculated to be $1.8 \times 10^{-11} \text{ cm}^2 \text{ s}^{-1}$ at room temperature (**Supplementary Fig. S32**). Typical polymers for photoconversion application, the exciton diffusion and charge transfer dynamics of poly(3-hexyl thiophene) (P3HT) and phenyl-C61-buryric acid methyl ester (PCBM) have been studied, giving the diffusion coefficient of 1.8×10^{-3} and $2.7 \times 10^{-4} \text{ cm}^2 \text{ s}^{-1}$, respectively (*Adv. Mater.* **2008**, 20, 3516; *Nanoscale* **2011**, 3, 2280). The diffusion of charge carrier is much faster than that of the adsorbed gas by several orders of magnitude,

implying that the gas diffusion should have more crucial influence at the rate limiting rather than the charge carriers.

In the present study, we aim to provide new insights into the design and synthesis of well-defined porous photocatalysts for CO₂ uptake and conversion. The designed sandwiched structure is somewhat complicated and makes it impossible to establish individual kinetic models. Although the kinetic mechanism cannot be clearly clarified in the current study, we indeed have made much progress on the kinetic analysis of photocatalytic CO₂ conversion that is seldom discussed in the literature. Inspired by the reviewers' suggestions, we have realized that the kinetic analysis is indeed very important for photocatalytic CO₂ conversion and there are yet many challenges that need to be addressed. In future study, we will use the simple photocatalytic system such as pure TiO₂ photocatalyst to probe the kinetics model and reaction mechanism of CO₂ conversion, and then extend to the complicated models involving adsorption, diffusion and photocatalytic processes.

On pages 12-13, all results and discussions on kinetic analysis are included and displayed as a separate section "kinetic analysis" in the revised manuscript.

“Kinetic analysis. The kinetics experiments were carried out to understand the contribution of CO₂ adsorption and diffusion to the enhancement of photocatalytic efficiency. The relationship between the CO₂ adsorption and CH₄ production can be explored by varying the surface coverage of CO₂ on the active sites. The partial pressure of CO₂ is adjusted in CO₂/N₂ mixture because of a high CO₂/N₂ selectivity ratio of 25.8 over the HCP-TiO₂-FG photocatalyst (**Supplementary Fig. S29**). Since the kinetic model and reaction mechanism of photocatalytic CO₂ conversion are ambiguous so far, the quantitative relationship between CO₂ coverage and CH₄ evolution rate is still unclear. Interestingly, it is observed that they show a similar trend of increase with CO₂ proportion, e.g. both of them dramatically increased at lower partial pressure and then displayed a slow increase at higher CO₂ concentration (**Supplementary Fig. S30**). Generally, the reaction rates that are normalized to the active sites allow the direct comparison of intrinsic reactivity on different catalysts⁴⁷⁻⁴⁹. For the catalytic system employing same catalyst, the reaction rate appears to be independent of the loading amount of catalyst after normalization to the same amount^{50, 51}. In this regard, the porous HCP-TiO₂-FG photocatalyst possesses equivalent catalytic active sites to TiO₂/HCP-FG hybrid due to the same content of TiO₂ photocatalyst. That is, the more efficient CH₄ production over HCP-TiO₂-FG should not result from the difference in the number of catalytic sites but mostly come from the higher surface coverage of CO₂ on the active sites.

The temperature has a complicated influence on the rate of photocatalytic conversion from the aspects of adsorption and diffusion. By increasing the temperature, the surface coverage of CO₂ molecules on the catalyst surface was decreased due to the exothermic effect of adsorption process (**Fig. 3e-f**), while the diffusion rate was increased as a result of the increased thermal motion of

CO₂ molecules (**Supplementary Fig. S31**). Based on Arrhenius plot, the adsorption activation energy for CO₂ adsorption is calculated to be 5.20 kJ mol⁻¹ (**Supplementary Fig. S32a**) using a microporous diffusion model^{52, 53}. Since the CH₄ production increases linearly and possesses dominant electron consumption selectivity as 83.7%, we can use the pseudo-zero order model to estimate the rate constant for the overall reaction, obtaining apparent activation energy of 9.34 kJ mol⁻¹ (**Supplementary Fig. S32b**). The diffusion process was further studied by varying the stirring speed. As shown in **Supplementary Fig. S33**, the increase of stirring speed greatly facilitates the photocatalytic conversion of CO₂ to CH₄ product. Combining the diffusion effect with pressure-/temperature- dependent characteristics, we can conclude that the photocatalytic CO₂ reduction over HCP-TiO₂-FG is not under intrinsic kinetic control of the catalyst but the efficiency is rather determined by gas adsorption and diffusion. The elucidation of adsorption and diffusion that contributed to the photocatalytic reaction, and is seldom discussed in the literature, provides valuable information for understanding the relationship between the catalytic performance and structure properties. As a result, it clearly demonstrates the superiority of such porous sandwich structure towards the visible-light-driven photocatalytic CO₂ conversion. Further kinetic study is required to probe the kinetics model and reaction mechanism of photocatalytic CO₂ conversion.”

Figure S29. CO₂ and N₂ adsorption isotherms of porous HCP-TiO₂-FG sandwich structure at 273 K. The HCP-TiO₂-FG photocatalyst exhibits a high CO₂/N₂ selectivity ratio of 25.8 calculated by the initial slopes of adsorption isotherms³⁷.

Figure S30. Influence of the partial pressure of CO₂ on the CO₂ uptake (a) and CH₄ production rate (b). The photocatalytic reactions were carried out in a batch system under standard atmospheric pressure. The partial pressure of CO₂ can be adjusted from 2.5% to 100% by varying the volume ratio of CO₂ to N₂. q_e is the equilibrium adsorption capacity at pure CO₂ atmosphere with 1 bar. q/q_e represents fractional uptake at different partial pressure of CO₂.

Figure S31. (a, b, c, d) Adsorption kinetic curves of HCP-TiO₂-FG at different temperature. (e) Fractional adsorption uptake (q_t/q_e) at different temperature. (f) Plots of the fractional adsorption uptake (q_t/q_e) against the square root of adsorption time at different temperature.

Figure S32. Arrhenius plot of CO₂ diffusivity (a) and CH₄ production rate (b) over HCP-TiO₂-FG sandwich structure. The diffusion coefficient D_M is calculated using a microporous diffusion model: $\frac{q_t}{q_e} \cong \frac{6}{r_c} \sqrt{\frac{D_M}{\pi}} \sqrt{t}$, r_c is the average particle size³⁹.

Figure S33. Effect of stirring speed on CH₄ and CO production over the HCP-TiO₂-FG photocatalyst under visible-light ($\lambda \geq 420$ nm) irradiation.

The authors made an effort to identify the corresponding oxidation Reaction which has to balance the observed reduction reactions. Oxygen has been proven but quantitative Analysis is not yet available. What about oxidation of the formed products, e.g. CO and CH₄ to CO₂ as reverse reactions of the attempted CO₂ reduction. I suggest reference Experiments with these Substrates. Overall, the contribution may become suitable for Nature Communications after providing further evidence and comments to the points made before.

Response: It is reported that the evolved H₂ and O₂ quickly diminish over Pt loaded g-C₃N₄ photocatalyst after turning off the Xe lamp for one hour, indicating the fast occurrence of the backward reaction of water splitting on the Pt species (*Chem. Sci.* **2016**, 7, 3062). Following the reviewers' suggestion, we studied the reverse reactions of CO₂ reduction over HCP-TiO₂-FG photocatalyst. As shown in supplementary information (**Supplementary Fig. S25**), the amount of CO and CH₄ remained almost constant when the light was turned off. The extremely slow rate of reverse reaction indicates that the oxidation of CO and CH₄ to CO₂ is efficiently controlled over the HCP-TiO₂-FG photocatalyst under such mild reaction conditions.

In the revised manuscript, we have added the following text on page 11:

“When the light was turned off, the amount of CO and CH₄ remained almost constant, indicating the efficient control over the reverse reactions of CO₂ reduction (oxidizing CO and CH₄ to CO₂) over the HCP-TiO₂-FG photocatalyst under such mild reaction conditions (Supplementary Fig. S25).”

Figure S25. Changes in CH₄ and CO production over the HCP-TiO₂-FG photocatalyst under dark conditions and visible-light ($\lambda \geq 420$ nm) irradiation and dark conditions.

Reviewers' comments:

Reviewer #2 (Remarks to the Author):

Comments:

I am pleased again to see that the authors have made some efforts to improve the quality of this manuscript by performing new experiments, adding new discussion and completing a number of changes. Indeed, some comments have been partially addressed. However, the key issue in terms of structure-performance relationship is still ambiguous, which makes it insufficient to meet the level for publication in Nature Commun. For this reason, I have to recommend the rejection toward the publication in Nature Commun.

Specific comments are listed below:

1. By TEM images and EDX mapping, the authors demonstrated in the current revised manuscript that a distinct sandwich structure of HCP-TiO₂-FG, in which the graphene surface and TiO₂ crystals were covered by the HCPs layers and the TiO₂ crystals were supported on the graphene sheets and encapsulated by the ultrathin HCPs layer, was formed. However, in the Response Letter, the authors pointed out that the HCPs and graphene existed as an integrated structure instead of freestanding parts in the composite. This seems to be very ambiguous.

2. According to Fig. 3d, the TiO₂ intercalation somewhat restricted the aggregation of HCPs layers on graphene nanosheets and thus dramatically enlarged the specific surface area. As a result, the enhanced performance of HCP-TiO₂-FG cannot support the argument that the catalytic sites are located on TiO₂. As the HCP-FG material also exhibits a photocatalytic activity for CO and CH₄ production, it remains elusive whether the catalytic sites are located on TiO₂ or HCP-FG.

Reviewer #3 (Remarks to the Author):

The careful revision of the authors is acknowledged. With regard to kinetic analysis and mass Transfer, all parameters reasonable well accessible by experiments were investigated and a careful discussion was added. The presented additional data appear comprehensive. I support acceptance.

Response to Reviewer' Comments

Many thanks for forwarding us the reviewer' comments that have helped us to further improve the manuscript. Given below are our responses (in BLUE colour) to the editor's queries.

Reviewer 2:

1. By TEM images and EDX mapping, the authors demonstrated in the current revised manuscript that a distinct sandwich structure of HCP-TiO₂-FG, in which the graphene surface and TiO₂ crystals were covered by the HCPs layers and the TiO₂ crystals were supported on the graphene sheets and encapsulated by the ultrathin HCPs layer, was formed. However, in the Response Letter, the authors pointed out that the HCPs and graphene existed as an integrated structure instead of freestanding parts in the composite. This seems to be very ambiguous.

Response: In fact, we did not mention the "integrated structure" in the manuscript but just used this description in the response letter of R2. The English dictionary translates "integration" as "combining two or more things so that they work together". Therefore we believe that the description of "sandwich structure" and "integrated structure" are not much contradictory. While responding to the comment 2# of Reviewer 2#, we used "integrated structure" in the response letter only to emphasize that the charge carriers can be moved throughout the whole HCP-FG structure.

Here we would like to explain the relationship between HCPs and graphene as HCPs being covalently linked on graphene through the methylene linkers. The TiO₂ crystals were intercalated into the HCPs layers and graphene sheets at discrete sites, as shown in **Fig. 1**. Therefore, the "HCP-TiO₂-FG sandwich structure" is used to describe these sites with TiO₂ crystals loading. At the other sites without TiO₂ crystals, the HCPs are covalently linked on graphene that seems to be a "HCP-FG integrated structure". The model of covalent linking is now also presented in **Fig. 1** according to the structural characterizations.

2. According to Fig. 3d, the TiO₂ intercalation somewhat restricted the aggregation of HCPs layers on graphene nanosheets and thus dramatically enlarged the specific

surface area. As a result, the enhanced performance of HCP-TiO₂-FG cannot support the argument that the catalytic sites are located on TiO₂. As the HCP-FG material also exhibits a photocatalytic activity for CO and CH₄ production, it remains elusive whether the catalytic sites are located on TiO₂ or HCP-FG.

Response: We have also explained the identification of catalytic sites in our previous response to R1 (response to the Comment 6# of Reviewer 2#). Given below is further explanation to clarify it.

Let's first compare the porous property and photocatalytic performance to clarify the location of catalytic sites. The results of HCP-FG, HCP-TiO₂-FG, and TiO₂/HCP-FG samples are taken from the manuscript and supplementary information, as shown in the **Fig. R1 and Table R1** below. Obviously, the introduction of porous HCPs layers enriched the adsorptive sites to achieve the high CO₂ uptake and improved the visible light absorption. Thus the formation of well-defined HCP-TiO₂-FG sandwich structure resulted in much higher photocatalytic CO₂ reduction rate. The HCP-FG material also exhibited broad visible-light absorption, high surface area and notable CO₂ uptake. However, its photocatalytic performance was far less than that of HCP-TiO₂-FG, especially in the eight-electron reduction to CH₄. These results show that the catalytic sites on TiO₂ are much more active for CO₂ reduction than those on HCP-FG. Besides, the catalytic sites can be further clarified by the comparison between HCP-FG and TiO₂/HCP-FG hybrid. When TiO₂ crystals were supported on HCP-FG to form TiO₂/HCP-FG hybrid, it is found that TiO₂ deposition blocked most of the adsorptive sites of HCP-FG and resulted in a dramatic decrease to less than one-third of CO₂ uptake. However, the CH₄ production over TiO₂/HCP-FG was 7.4 times more than that over pristine HCP-FG. These results confirm that the TiO₂ crystals are introduced as catalytically active sites to facilitate the photocatalytic CO₂ conversion.

Second, the pathway of charge carriers transfer and separation was studied to clarify the location of catalytic sites. As shown in **Fig. 4f**, the lowest unoccupied molecular orbital (LUMO) level of HCP-FG is more negative than the conduction band (CB) level of TiO₂. It is well-known that the polymer materials usually possess the excitons with high binding energy, which usually recombine at the excited states. The photogenerated electrons of the excited HCP-FG can migrate to the CB of TiO₂ due to their matched energy levels (Type II heterojunction model: *Semiconductors*

1998, 32, 1; *Angew. Chem., Int. Ed.* **2012**, 51, 10145; *ACS Catal.* 2014, 4, 3637; *Energy Environ. Sci.* **2015**, 8, 731; *Adv. Mater.* **2017**, 29, 1606198). Thus the photogenerated carriers of HCP-FG can be separated at the interface with TiO₂, which largely reduced the recombination loss. Since the electrons are located on TiO₂ crystals, the adsorbed CO₂ molecules are more readily converted to CO than CH₄ at the catalytic sites of TiO₂.

Based on the discussion above, it can be inferred that the CO₂ reduction better achieved at the catalytic sites on TiO₂ rather than those on HCP-FG. The identification of catalytic sites could better explain the difference in photocatalytic performance among HCP-FG, HCP-TiO₂-FG and TiO₂/HCP-FG.

We have also mentioned the above discussion at page 9 and 11 of the manuscript. Now we have also made further revision to better clarify the location of catalytic sites:

On page 9, “The comparison among HCP-FG, HCP-TiO₂-FG and TiO₂/HCP-FG shows that the catalytic sites on TiO₂ are much more active for CO₂ reduction than those on HCP-FG.”

On page 11, “The photogenerated electrons of the excited HCP-FG can migrate to the CB of TiO₂ via their interfacial interaction. Thus the CO₂ reduction is better achieved at the catalytic sites on TiO₂ rather than those on HCP-FG, which is in consistent with the above discussion on the porous property and photocatalytic performance.”

Figure R1. (a) Nitrogen adsorption and desorption isotherms at 77.3 K of samples. (b) Volumetric

CO₂ adsorption isotherms and desorption isotherms up to 1.00 bar at 273.15 K of samples. (c) Pore size distribution that calculated using DFT methods (slit pore models, differential pore volumes). Time-dependent production of CH₄ (d) and CO (e) in photocatalytic CO₂ reduction with different catalysts under visible-light ($\lambda \geq 420$ nm). (f) Average efficiency of photocatalytic CO₂ conversion with different catalysts during 5 h of visible-light ($\lambda \geq 420$ nm) irradiation.

Table R1. Comparison of the selected samples in porous property and photocatalytic performance.

Sample	S _{BET} ^a (m ² g ⁻¹)	CO ₂ uptake ^b (wt %)	PV ^c (cm ³ g ⁻¹)	Visible-light irradiation ($\mu\text{mol g}^{-1} \text{h}^{-1}$)		
				r(CH ₄) ^d	r(CO) ^e	R _e ^f
HCP-FG	593	10.90	0.380	0.90	7.62	22
TiO ₂ /HCP-FG	178	3.31	0.164	6.71	16.17	86
HCP-TiO ₂ -FG	988	12.87	0.693	27.62	61.63	264

^a Surface area calculated from nitrogen adsorption isotherms at 77.3 K using BET equation. ^b CO₂ uptake determined volumetrically using a Micromeritics ASAP 2020 M analyzer at 1.00 bar and 273.15 K. ^c Pore volume calculated from nitrogen isotherm at P/P₀=0.995, 77.3 K. ^{d,e} average of gas evolution rate (r) during 5 h of photocatalytic CO₂ reduction. ^f R_e is the rate of total consumed electron number for the reduced product; R_e = 8r(CH₄)+2r(CO).

Reviewers' comments:

Reviewer #2 (Remarks to the Author):

I still think that the key issue of structure-performance relationship remains ambiguous, which makes the manuscript insufficient to meet the level for publication in Nature Commun. If the authors could address the comments, I would recommend it for publication.

Specific comments are listed below:

1. The advantage of the "sandwich structure" is still not clear. As described by the authors, the catalytic sites are located on TiO₂ while the adsorptive sites are on HCP, and the graphene improve the photogenerated electrons transfer from HCP to TiO₂. In this case, the best structure may be TiO₂-FG-HCP rather than HCP-TiO₂-FG.
2. The authors demonstrated that the photogenerated carriers of HCP-FG can be separated at the interface with TiO₂. However, according to the "sandwich structure", there is no interface between HCP-FG and TiO₂. Instead, the interface should be formed between FG and TiO₂ or HCP and TiO₂. In addition, according to the "HCP-TiO₂-FG sandwich structure", the TiO₂ intercalation somewhat restricted the aggregation of HCPs layers on graphene nanosheets, but also blocked the charge transfer from HCP to TiO₂ through graphene.
3. In the "sandwich structure", the catalytic sites are located on TiO₂ while the adsorptive sites are on HCP, as stated by the authors. This requires a short diffusion length for the transfer of CO₂ molecules from adsorptive sites to catalytic sites. However, there is no direct evidence to support this argument.

Response to Reviewer' Comments

Given below are our responses (in BLUE colour) to the reviewer' comments. The changes to the manuscript and supplementary information are marked in RED colour.

Reviewer 2:

I still think that the key issue of structure-performance relationship remains ambiguous, which makes the manuscript insufficient to meet the level for publication in Nature Commun. If the authors could address the comments, I would recommend it for publication.

Response: We thank the reviewer for his/her continuous interest in our work. According to the comments, we have further clarified the proposed HCP-TiO₂-FG structure and the structure-performance relationship by supplementing the related experiments and discussions. We have now addressed all the reviewer's queries point-wise in our response below.

1. The advantage of the "sandwich structure" is still not clear. As described by the authors, the catalytic sites are located on TiO₂ while the adsorptive sites are on HCP, and the graphene improve the photogenerated electrons transfer from HCP to TiO₂. In this case, the best structure may be TiO₂-FG-HCP rather than HCP-TiO₂-FG.

Response: We realized that there might still be misunderstanding of the structure which caused confusion to the reviewer, so we have further clarified the structure in the revised version. The TiO₂ crystals were supported on the graphene sheets and then encapsulated by the ultrathin HCPs layers after knitting *syn*-PhPh₃ with functionalized graphene. The design of such porous HCP-TiO₂-FG photocatalysts aimed to combine the TiO₂ photocatalyst with catalytically active sites, hypercrosslinked polymer with high CO₂ uptake and graphene with the high charge mobility. The proposed structure may not be a strict "sandwich structure". Since the TiO₂ crystals were intercalated into the HCPs layers and graphene sheets at discrete sites, we previously used the "HCP-TiO₂-FG sandwich structure" to describe these

sites with TiO_2 crystals loading. Considering that this name caused confusion to the reviewer, we have replaced it by the “porous HCP- TiO_2 -FG composite”. For better understanding, we have now revised the models in the diagram to show the interface between TiO_2 and HCP-FG (**Fig. 1**). As you will see, the photogenerated electron-hole pairs of the excited HCP-FG can move throughout the HCP-FG structure and then be separated at the interface with TiO_2 via their interfacial interaction, as depicted in **Supplementary Fig. S31**.

Figure 1. Construction of a well-defined porous HCP- TiO_2 -FG composite structure. (I) The functionalization of TiO_2 -G by diazonium salt formation. (II) The knitting of TiO_2 -FG with *syn*-PhPh₃ by solvent knitting method. The magnified model in the top right corner is the cross-section of HCP- TiO_2 -FG composite.

Figure S31. Diagram of charge separation at the interface of HCP-FG with TiO_2 .

The proposed structure was previously demonstrated by TEM, STEM, HRTEM, SEM and AFM observations. In order to display the spatial distribution of HCP layers, we conducted further characterization by high-angle annular dark field (HAADF) mapping and three-dimensional rotating techniques. The results are now presented in

Fig. 2f-i, **Supplementary Fig. S6** and **Supplementary Video**, which could verify the above porous HCP-TiO₂-FG composite structure. The detailed descriptions of the structure have been added in the morphology characterization section. We are hoping that the structure is now much clear and clearly understandable.

As suggested, another type of composite, HCP-FG supported TiO₂ (TiO₂/HCP-FG), has already been prepared for comparison in this work. The HCP layers were hypercrosslinked on the functionalized graphene to form the HCP-FG structure with graphene surface rarely exposed, and then the HCP-FG was used as a supporting material for TiO₂ crystals growth during the solvothermal process. For better understanding of structure, we present the models in the diagram below (**Fig. R1**). Model (a) shows HCP-TiO₂-FG structure. Model (b) is the composite of supporting TiO₂ on HCP-FG, which is named as TiO₂/HCP-FG in our manuscript. The liquid-phase synthesis in this work could not restrict the location of TiO₂ crystals on the side of graphene, which are different from the fabrication of layered structure by layer-by-layer assembly or spin-coating technique. In this regard, the comparison with TiO₂/HCP-FG is a better way to better explain the superior nature of the designed HCP-TiO₂-FG structure. The morphology of the TiO₂/HCP-FG composite is presented in **Supplementary Fig. S13**. The results of HCP-FG, HCP-TiO₂-FG and TiO₂/HCP-FG are taken from the manuscript and supplementary information, as shown in the **Fig. R2** and **Table R1** below.

Figure R1. (I) Formation of HCP-TiO₂-FG structure (Model (a)) by knitting of TiO₂-FG with syn-PhPh₃. (II) Formation of TiO₂/HCP-FG structure (Model (b)) by supporting TiO₂ on HCP-FG.

Based on the above discussions, when TiO_2 was supported on HCP-FG instead of graphene, the TiO_2 deposition blocked most of the adsorptive sites of HCP-FG and resulted in a dramatic decrease in surface area, CO_2 uptake and photocatalytic efficiency. As a result, the designed HCP- TiO_2 -FG composite structure appears to be much superior to the TiO_2 /HCP-FG composite in our system. Of course, we expect that, in future, the synthesis strategy can be further improved to yield a much better structured combination of microporous organic polymers with photocatalysts.

We have also mentioned the above discussions in the revised manuscript and supplementary information.

On Pages 4, *“The elemental mapping images in **Fig. 2f-i** clearly display the thin HCP shells wrapping the surface of TiO_2 crystals. By rotating the angle of the sample, multiple images were collected to create a three-dimensional TEM (3D-TEM) video (**Supplementary Fig. S6 and Supplementary Video**) to further elucidate the HCP- TiO_2 -FG composite structure with distinct interface between TiO_2 and HCP-FG. Based on the above analysis, it can be deduced that the TiO_2 crystals were supported on the graphene sheets and then encapsulated by the ultrathin HCPs layers after knitting syn-PhPh₃ with functionalized graphene, as shown in **Fig. 1.**”*

On Page 7, *“To further understand the effect of such well-defined HCP- TiO_2 -FG composite structure on improving surface area and CO_2 uptake, another type of composite, TiO_2 /HCP-FG, was prepared as a control by changing the order of introducing HCP and TiO_2 . The HCP layers were hypercrosslinked on the functionalized graphene to form HCP-FG at first, and then TiO_2 crystals were grown on the HCP-FG surface during the solvothermal process. Owing to the high surface area and porous property of HCP-FG support, the TiO_2 particles of TiO_2 /HCP-FG possessed much smaller size than that of TiO_2 -G or HCP- TiO_2 -FG (**Supplementary Fig. S13**). However, the hypercrosslinking reaction caused the graphene surface to be almost entirely covered by HCP layers, so that most of TiO_2 crystals were assembled on the HCP surface rather than be encapsulated by HCPs layers like those in HCP- TiO_2 -FG composite. The results implied the interaction of TiO_2 with graphene*

serving as “bridge” to the formation of well-defined HCP-TiO₂-FG composite structure. The TiO₂/HCP-FG composite showed a high surface area of 178 m² g⁻¹ and CO₂ uptake of 3.31 wt% relative to TiO₂ and TiO₂-G (**Supplementary Fig. S14**), but much lower than that of HCP-TiO₂-FG. According to the previous such reports, the incorporation of semiconductor photocatalysts generally decrease the surface area and CO₂ uptake of the capture materials, mainly resulting from the semiconductors with low surface area occupying the porous surface^{19, 23, 36, 37}. Hence the superiority of HCP-TiO₂-FG in CO₂ adsorption could be ascribed to the well-defined porous composite structure with TiO₂ encapsulated inside the HCP-FG network instead of being assembled on the surface”

On Page 9, “To further elucidate the superiority of **HCP-TiO₂-FG composite**, the CO₂ conversion efficiency over **TiO₂/HCP-FG composite** was evaluated. It was found that the photocatalytic activity of TiO₂/HCP-FG hybrid was much lower than that of HCP-TiO₂-FG sandwich structure. It may be due to the blockage of the porous structure of HCP-FG with TiO₂ crystals thereby decreasing the surface area and CO₂ uptake and subsequently leading to the reduced CO₂ reduction rate. Although TiO₂ deposition blocked most of the adsorptive sites of HCP-FG and resulted in a dramatic decrease to less than one-third of CO₂ uptake, the CH₄ production over TiO₂/HCP-FG was, however, 7.4 times more than that over pristine HCP-FG. The comparison among HCP-FG, HCP-TiO₂-FG and TiO₂/HCP-FG shows that the catalytic sites on TiO₂ are much more active for CO₂ reduction than those on HCP-FG. Based on the above analysis, it can be deduced that the in-situ knitting strategy for HCP-TiO₂-FG can effectively produce **porous structure** without significant pore blockage of the porous polymers.”

On Pages 2-3 **Supplementary information**, “**The high resolution transmission electron microscopy (HR-TEM) and scanning transmission electron microscopy (STEM) images of samples were recorded on a Tecnai G2 F30 microscope (FEI Corp. Holland). The High-Angle Annular Dark Field (HAADF) mapping images and three-dimensional TEM (3D-TEM) video were obtained on a Talos F200X field-emission transmission electron microscope. The 3D-TEM video in the**

Supplementary Video was created by taking multiple views of the sample at differing angles from -50° to 50° with 2° increments.”

On Pages 9 **Supplementary information**, “By rotating the angle of the sample, multiple images were obtained to create a 3D-TEM video (Supplementary video) and a tilt series of images at the selected angles are displayed in Supplementary Fig. S6. The TiO_2 crystals in the labeled area can be distinctly observed at the angles of around 0° (d-f). By rotating the angle far from 0° , they were gradually hidden inside the HCP layers, which could be identified by the wrinkles of the HCP outer layer (a-c and g-h). Therefore, the TiO_2 crystals on the graphene sheets were not exposed outside but encapsulated by the ultrathin HCPs layer.”

Figure 2. Morphology and elemental mapping of various photocatalysts. TEM images of TiO_2 -G (a, b) and HCP- TiO_2 -FG (c, d) at different magnification. The insets in b and d are the corresponding HR-TEM images. STEM image of HCP- TiO_2 -FG (e). (f-i) High-angle annular dark field (HAADF) mapping images of HCP- TiO_2 -FG.

Figure S6. Multiple views of the HCP-TiO₂-FG sample at the selected angles of (a) -50°, (b) -38°, (c) -24°, (d) -10°, (e) 0°, (f) 10°, (g) 24°, (h) 38°. The insets are the rotation of the selected two crystals that labeled by the white arrow in (e).

Figure S13. TEM images of HCP-FG supported TiO₂ (TiO₂/HCP-FG) at different magnification.

Figure R2. (a) Nitrogen adsorption and desorption isotherms at 77.3 K of samples. (b) Volumetric CO₂ adsorption isotherms and desorption isotherms up to 1.00 bar at 273.15 K of samples. (c) Pore size distribution that calculated using DFT methods (slit pore models, differential pore volumes). Time-dependent production of CH₄ (d) and CO (e) in photocatalytic CO₂ reduction with different catalysts under visible-light ($\lambda \geq 420$ nm). (f) Average efficiency of photocatalytic CO₂ conversion with different catalysts during 5 h of visible-light ($\lambda \geq 420$ nm) irradiation.

Table R1. Comparison of the selected samples in porous property and photocatalytic performance.

Sample	$S_{\text{BET}}^{\text{a}}$ (m ² g ⁻¹)	CO ₂ uptake ^b (wt %)	PV ^c (cm ³ g ⁻¹)	Visible-light irradiation (μmol g ⁻¹ h ⁻¹)		
				$r(\text{CH}_4)^{\text{d}}$	$r(\text{CO})^{\text{e}}$	R_e^{f}
HCP-FG	593	10.90	0.380	0.90	7.62	22
TiO ₂ /HCP-FG	178	3.31	0.164	6.71	16.17	86
HCP-TiO ₂ -FG	988	12.87	0.693	27.62	61.63	264

^a Surface area calculated from nitrogen adsorption isotherms at 77.3 K using BET equation. ^b CO₂ uptake determined volumetrically using a Micromeritics ASAP 2020 M analyzer at 1.00 bar and 273.15 K. ^c Pore volume calculated from nitrogen isotherm at P/P₀=0.995, 77.3 K. ^{d,e} average of gas evolution rate (r) during 5 h of photocatalytic CO₂ reduction. ^f R_e is the rate of total consumed electron number for the reduced product; $R_e = 8r(\text{CH}_4) + 2r(\text{CO})$.

2. The authors demonstrated that the photogenerated carriers of HCP-FG can be separated at the interface with TiO₂. However, according to the “sandwich structure”, there is no interface between HCP-FG and TiO₂. Instead, the interface should be formed between FG and TiO₂ or HCP and TiO₂. In addition, according to the

“HCP-TiO₂-FG sandwich structure”, the TiO₂ intercalation somewhat restricted the aggregation of HCPs layers on graphene nanosheets, but also blocked the charge transfer from HCP to TiO₂ through graphene.

Response: Since the proposed structure of HCP-TiO₂-FG has been clarified in the above response, we are hoping that the interface between TiO₂ and HCP-FG could be clearly understandable now. As shown in **Fig. 1**, the TiO₂-FG interface was first formed that served as “bridge” to favor the formation of interface between TiO₂ and HCP-FG. It should be noted that the HCP was covalently linked with FG for the formation of HCP-FG integrated structure instead of freestanding HCP blocks. Hence the interface we observed in **Fig. 1-2** was actually the interface between TiO₂ and HCP-FG. In the absence of TiO₂, the surface area and CO₂ uptake of HCP-FG were much lower than those of HCP-TiO₂-FG (**Supplementary Tab. S2**). It was also found that the HCP-FG layers of TiO₂/HCP-FG were much thicker than those of HCP-TiO₂-FG (**Fig. 2** and **Supplementary Fig. S13**). The results implied that the TiO₂ intercalation somewhat restricted the aggregation of HCPs layers on graphene nanosheets. According to the charge separation pathway in **Figure 4f**, the photogenerated electron-hole pairs of the excited HCP-FG can move throughout the HCP-FG structure and then be separated at the interface with TiO₂ *via* their interfacial interaction. In the prerequisite of better understanding the structure, we can conclude that the graphene plays dual a role in the composite photocatalyst: (1) the interaction with TiO₂ serving as “bridge” to form well-defined porous HCP-TiO₂-FG composite structure; (2) the covalent linking with HCPs improving the electronic conductivity of the HCPs and thus facilitating the electron transfer from HCP-FG to TiO₂ in the composite. For better understanding, here we would like to present a simple diagram illustrating the charge separation at the interface (**Supplementary Fig. S31**). We believe that given the above discussion, the query regarding “TiO₂ blocked the charge transfer from HCP to TiO₂ through graphene” is now better addressed.

Given below is the related explanation in the manuscript.

On Page 7, *“The results implied the interaction of TiO₂ with graphene serving as “bridge” to the formation of well-defined HCP-TiO₂-FG composite structure.*

On Page 11, *“the covalent linking with graphene effectively improves the electronic conductivity of the HCPs and thus facilitates the electron transfer in the composite.*

The less efficient CH_4 production over HCP- TiO_2 photocatalyst can also reflect the influence of graphene on improving the charge separation efficiency (**Supplementary Fig. S27**)... Under visible-light irradiation, HCP-FG functions both as CO_2 adsorbent and photosensitizer, which directly absorbs the photons to induce the HOMO to LUMO transition. The photogenerated *electron-hole pairs* of the excited HCP-FG can move throughout the HCP-FG structure and separated at the interface with TiO_2 via their interfacial interaction, as shown in **Supplementary Fig. S31**. The photogenerated electrons of the excited HCP-FG can migrate to the CB of TiO_2 via their interfacial interaction.”

Figure S31. Diagram of charge separation at the interface of HCP-FG with TiO_2 .

3. In the “sandwich structure”, the catalytic sites are located on TiO_2 while the adsorptive sites are on HCP, as stated by the authors. This requires a short diffusion length for the transfer of CO_2 molecules from adsorptive sites to catalytic sites. However, there is no direct evidence to support this argument.

Response: In the HCP- TiO_2 -FG composite, the HCP outer layers serve as adsorptive sites for CO_2 molecules and the TiO_2 crystals function as catalytic sites for CO_2 reduction. We agree that the diffusion of CO_2 molecules from the adsorptive sites to the catalytic sites plays an important role in CO_2 conversion.

(1) The evidence on the role of diffusion

The kinetic study of CO_2 adsorption and diffusion provided a convincing evidence to clarify the significant role of diffusion in such a gas-solid reaction system. The related discussion is available at pages 12-13. More detailed descriptions can also be found in the response letter of R2 (response to the Comment of Reviewer #3). In the following peer-review process, the Reviewer #3 made comments on our revision as “The careful revision of the authors is acknowledged. With regard to kinetic analysis and mass transfer, all parameters reasonable well accessible by experiments were investigated and a careful discussion was added. The presented additional data appear

comprehensive.” For your convenience, we have pasted the previous response at the bottom as reference.

(2) The evidence on the effect of diffusion length

Since the gas diffusion has a crucial influence on the rate limiting for CO₂ conversion, an appropriate diffusion length should be beneficial for the diffusion of CO₂ molecules from the adsorptive sites to the catalytically active sites. According to the structural analysis, the TiO₂ crystals were encapsulated by ultrathin HCP outer layers with a thickness of 3~8 nm (**Fig. 2**), indicating a short diffusion length for CO₂ molecules diffusing from HCP-FG to TiO₂ photocatalysts. For better understanding, we present the CO₂ diffusion model in the diagram below (**Supplementary Fig. S15**).

Figure S15. Diagram of CO₂ diffusion from adsorptive sites to catalytic sites for conversion.

In addition, we have prepared the HCP-TiO₂-FG composite with different thickness of HCP outer layers to study the effect of the diffusion length on the CO₂ uptake and conversion efficiency. The results suggested that there is an appropriate thickness of HCP layers which balanced the CO₂ adsorption and diffusion. In this work, we tried to construct the HCP-TiO₂-FG sandwich structure with ultrathin HCP layers to reduce the diffusion length. Of course, we expect that, in the future, the synthesis strategy can be further improved to yield a much better structure with more efficient CO₂ adsorption and diffusion.

Given below is the related explanation in the manuscript.

On Page 9, “*The model of CO₂ diffusion and conversion was presented in Supplementary Fig. S15.*”

On Page 9-10, “*the HCPs obtained by this strategy are comprised of ultrathin layers with a thickness of 3~8 nm wrapping around TiO₂ crystals (Fig. 2d-f), which facilitates the diffusion of CO₂ molecules from the adsorptive sites on HCPs layers to the catalytic sites on TiO₂ photocatalysts. The effect of the thickness of HCP layers on*

the CO₂ conversion efficiency was studied by adjusting the amount of syn-PhPh₃. By increasing the amount of syn-PhPh₃, the mass ratio of TiO₂ was slightly decreased from 31% to 29% (**Supplementary Fig. S16**), however the size of TiO₂ particles was decreased accompanied by the thickening of the HCP layers (**Supplementary Fig. S17**), which suggests that the HCP outer layers effectively suppress the growth of TiO₂ crystals. The distinct thickening of the outer layers was further verified from the characteristic morphology revealed in Fig. 1, showing HCP layers being hypercrosslinked on FG surface and encapsulating TiO₂ crystals. The surface area and CO₂ uptake capacity increased with the amount of syn-PhPh₃ (**Supplementary Fig. S18a and Tab. S4**), on the other hand, the diffusion length of CO₂ molecules also increased due to the thickening of the outer layer. As the CO₂ conversion efficiency increased initially and then decreased at higher amount (**Supplementary Fig. S18b**), there may be an appropriate thickness of HCP layers that balance the CO₂ adsorption and diffusion.”

Figure S16. Thermogravimetric analysis of HCP-TiO₂-FG, HCP-TiO₂-FG-1 and HCP-TiO₂-FG-2 at heating rate of 10 °C /min under air. The HCP-TiO₂-FG, HCP-TiO₂-FG-1 and HCP-TiO₂-FG-2 were synthesized by adding 20, 25 and 30 mg of syn-PhPh₃, respectively.

Figure S17. TEM images of HCP-TiO₂-FG-1 (a, b) and HCP-TiO₂-FG-2 (c, d). The HCP-TiO₂-FG-1 and HCP-TiO₂-FG-2 were synthesized by adding 25 and 30 mg of syn-PhPh₃, respectively.

Supplementary Fig. S18. (a) Effect of syn-PhPh₃ amount on the CO₂ uptake. (b) Effect of syn-PhPh₃ amount on the CH₄ and CO production rates over HCP-TiO₂-FG composite photocatalysts under visible-light ($\lambda \geq 420$ nm).

Table S4 The porous property and CO₂ conversion efficiency of the HCP-TiO₂-FG composites with different amount of syn-PhPh₃.

Sample	S_{BET}^a ($m^2 g^{-1}$)	CO ₂ uptake ^b (wt %)	PV ^c ($cm^3 g^{-1}$)	Visible-light irradiation ($\mu mol g^{-1} h^{-1}$)		
				$r(CH_4)^d$	$r(CO)^e$	R_e^f
HCP-TiO ₂ -FG	988	12.87	0.693	27.62	21.63	264
HCP-TiO ₂ -FG-1	1136	14.65	0.758	30.45	28.52	301
HCP-TiO ₂ -FG-2	1362	16.45	0.799	12.94	14.97	133

Reference:

Query on diffusion from Reviewer #3

The authors were asked to provide further evidence for the governing nature of their two main Arguments, namely the superior CO₂ adsorption as pre-requisite of high substrate concentration close to or on the active site and their hypothesis on the shortend diffusion lengths. Indeed, a suitable kinetic analysis has been carried out illustrating that the investigated photocatalytic CO₂ reduction is not under intrinsic kinetic control of the catalyst but the observed rates of CO and CH₄ formation are rather determined by some transport effects. The observed limited dependence of rate on temperature may hint towards diffusion control. I fully agree that kinetic analyses is a yet under-represented aspect in photo-catalysis. Though, the little rate dependence on temperature appears to point towards film diffusion effects. Has the stirring speed or the main particle size been varied? It remains unclear for me why the authors conclude on a reduced Diffusion length for the Optimum System, although they do not know which Diffusion is rate limiting: CO₂ from bulk to the film, through the film, in/on the porous material or Charge carrier diffusion?

Response: Many thanks for appreciating our efforts on the kinetic analysis presented in the previous revision. The photocatalytic reactions were carried out in a gas-solid batch system without stirring the gas mixture. According to the suggestion, we studied the diffusion process by varying the stirring speed from zero to maximum. As shown in Supplementary Fig. S40, the increase in stirring speed greatly facilitates the photocatalytic conversion of CO₂ to CH₄ product, indicating that the diffusion plays a significant role in such a gas-solid reaction system. The related results and discussions have been added as a separate section “kinetic analysis” in the revised manuscript.

The particle size is another factor affecting the diffusion process, especially for internal diffusion. We would like to mention that the HCP-TiO₂-FG composite produced through this strategy possesses the specific particle size. In the experience of optimizing synthesis conditions, TiO₂ crystals with larger or smaller size are inclined to form agglomerates, which cannot uniformly decorate on graphene or be

fully wrapped by ultrathin HCPs layers. In the current system, we cannot achieve the varied particle size while keeping the designed structure with similar surface area, CO₂ uptake capacity and the number of active sites.

In order to reduce the diffusion length, we tried to construct the HCP-TiO₂-FG sandwich structure with ultrathin HCP layers. The reduced diffusion length should be beneficial for the diffusion of CO₂ and photo-generated charges to the catalytically active sites, both of which may contribute to the improved CO₂ conversion efficiency. Based on the gas diffusivity measurement, the diffusion coefficient of CO₂ in the HCP-TiO₂-FG sandwich structure is calculated to be $1.8 \times 10^{-11} \text{ cm}^2 \text{ s}^{-1}$ at room temperature (Supplementary Fig. S39). Typical polymers for photoconversion application, the exciton diffusion and charge transfer dynamics of poly(3-hexyl thiophene) (P3HT) and phenyl-C61-butyric acid methyl ester (PCBM) have been studied, giving the diffusion coefficient of 1.8×10^{-3} and $2.7 \times 10^{-4} \text{ cm}^2 \text{ s}^{-1}$, respectively (*Adv. Mater.* 2008, 20, 3516; *Nanoscale* 2011, 3, 2280). The diffusion of charge carrier is much faster than that of the adsorbed gas by several orders of magnitude, implying that the gas diffusion should have more crucial influence at the rate limiting rather than the charge carriers.

In the present study, we aim to provide new insights into the design and synthesis of well-defined porous photocatalysts for CO₂ uptake and conversion. The designed sandwiched structure is somewhat complicated and makes it impossible to establish individual kinetic models. Although the kinetic mechanism cannot be clearly clarified in the current study, we indeed have made much progress on the kinetic analysis of photocatalytic CO₂ conversion that is seldom discussed in the literature. Inspired by the reviewers' suggestions, we have realized that the kinetic analysis is indeed very important for photocatalytic CO₂ conversion and there are yet many challenges that need to be addressed. In future study, we will use the simple photocatalytic system such as pure TiO₂ photocatalyst to probe the kinetics model and reaction mechanism of CO₂ conversion, and then extend to the complicated models involving adsorption, diffusion and photocatalytic processes.

On pages 12-13, all results and discussions on kinetic analysis are included and

displayed as a separate section “kinetic analysis” in the revised manuscript.

*“Kinetic analysis. The kinetics experiments were carried out to understand the contribution of CO₂ adsorption and diffusion to the enhancement of photocatalytic efficiency. The relationship between the CO₂ adsorption and CH₄ production can be explored by varying the surface coverage of CO₂ on the active sites. The partial pressure of CO₂ is adjusted in CO₂/N₂ mixture because of a high CO₂/N₂ selectivity ratio of 25.8 over the HCP-TiO₂-FG photocatalyst (**Supplementary Fig. S36**). Since the kinetic model and reaction mechanism of photocatalytic CO₂ conversion are ambiguous so far, the quantitative relationship between CO₂ coverage and CH₄ evolution rate is still unclear. Interestingly, it is observed that they show a similar trend of increase with CO₂ proportion, e.g. both of them dramatically increased at lower partial pressure and then displayed a slow increase at higher CO₂ concentration (**Supplementary Fig. S37**). Generally, the reaction rates that are normalized to the active sites allow the direct comparison of intrinsic reactivity on different catalysts⁴⁷⁻⁴⁹. For the catalytic system employing same catalyst, the reaction rate appears to be independent of the loading amount of catalyst after normalization to the same amount^{50, 51}. In this regard, the porous HCP-TiO₂-FG photocatalyst possesses equivalent catalytic active sites to TiO₂/HCP-FG due to the same content of TiO₂ photocatalyst. That is, the more efficient CH₄ production over HCP-TiO₂-FG should not result from the difference in the number of catalytic sites but mostly come from the higher surface coverage of CO₂ on the active sites.*

*The temperature has a complicated influence on the rate of photocatalytic conversion from the aspects of adsorption and diffusion. By increasing the temperature, the surface coverage of CO₂ molecules on the catalyst surface was decreased due to the exothermic effect of adsorption process (**Fig. 3e-f**), while the diffusion rate was increased as a result of the increased thermal motion of CO₂ molecules (**Supplementary Fig. S38**). Based on Arrhenius plot, the adsorption activation energy for CO₂ adsorption is calculated to be 5.20 kJ mol⁻¹ (**Supplementary Fig. S39a**) using a microporous diffusion model^{52, 53}. Since the CH₄ production increases linearly and possesses dominant electron consumption*

selectivity as 83.7%, we can use the pseudo-zero order model to estimate the rate constant for the overall reaction, obtaining apparent activation energy of 9.34 kJ mol^{-1} (Supplementary Fig. S39b). The diffusion process was further studied by varying the stirring speed. As shown in Supplementary Fig. S40, the increase of stirring speed greatly facilitates the photocatalytic conversion of CO_2 to CH_4 product. Combining the diffusion effect with pressure-/temperature- dependent characteristics, we can conclude that the photocatalytic CO_2 reduction over HCP- TiO_2 -FG is not under intrinsic kinetic control of the catalyst but the efficiency is rather determined by gas adsorption and diffusion. The elucidation of adsorption and diffusion that contributed to the photocatalytic reaction, and is seldom discussed in the literature, provides valuable information for understanding the relationship between the catalytic performance and structure properties. As a result, it clearly demonstrates the superiority of such porous HCP- TiO_2 -FG composite towards the visible-light-driven photocatalytic CO_2 conversion. Further kinetic study is required to probe the kinetics model and reaction mechanism of photocatalytic CO_2 conversion.”

Figure S36. CO_2 and N_2 adsorption isotherms of porous HCP- TiO_2 -FG sandwich structure at 273 K. The HCP- TiO_2 -FG photocatalyst exhibits a high CO_2/N_2 selectivity ratio of 25.8 calculated by the initial slopes of adsorption isotherms³⁷.

Figure S37. Influence of the partial pressure of CO_2 on the CO_2 uptake (a) and CH_4 production rate (b). The photocatalytic reactions were carried out in a batch system under standard atmospheric pressure. The partial pressure of CO_2 can be adjusted from 2.5% to 100% by varying the volume ratio of CO_2 to N_2 . q_e is the equilibrium adsorption capacity at pure CO_2 atmosphere with 1 bar. q/q_e represents fractional uptake at different partial pressure of CO_2 .

Figure S38. (a, b, c, d) Adsorption kinetic curves of HCP- TiO_2 -FG at different temperature. (e)

Fractional adsorption uptake (q_t/q_e) at different temperature. (f) Plots of the fractional adsorption uptake (q_t/q_e) against the square root of adsorption time at different temperature.

Figure S39. Arrhenius plot of CO₂ diffusivity (a) and CH₄ production rate (b) over HCP-TiO₂-FG sandwich structure. The diffusion coefficient D_M is calculated using a microporous

diffusion model: $\frac{q_t}{q_e} \cong \frac{6}{r_c} \sqrt{\frac{D_M}{\pi}} \sqrt{t}$, r_c is the average particle size³⁹.

Figure S40. Effect of stirring speed on CH₄ and CO production over the HCP-TiO₂-FG photocatalyst under visible-light ($\lambda \geq 420$ nm) irradiation.

REVIEWERS' COMMENTS:

Reviewer #2 (Remarks to the Author):

I recommend its publication as is.